# Technical note: An assessment of the relative contribution of the Soret effect to open water evaporation

Michael L. Roderick[1], Callum J. Shakespeare[1,2]

[1]Research School of Earth Sciences, Australian National University, Canberra, Australia, 2601.

[2]ARC Centre of Excellence for Climate Extremes, Australian National University, Canberra, Australia, 2601.

*Correspondence to*: Michael L. Roderick (michael.roderick@anu.edu.au)

**Abstract.** It is standard practice to assume that evaporation from open water depends on the gradient in water vapor concentration as per Fick's law. However, Fick's law is only true in an isothermal system. In general, we anticipate an additional mass flux due to the temperature gradient (in a non-isothermal system) and this is known as Soret diffusion or the

Soret effect. Here we evaluate the magnitude of the Soret effect and find that under exceptional circumstances the Soret effect may be as high as ~ 5% of the classical concentration-dependent mass ('Fickian') diffusion but it will usually be less than 2% of that same flux. Further we evaluate the magnitude of an additional advective (or 'Stefan') flux not usually considered in hydrologic studies that may be as high as ~5% of the concentration-dependent diffusion. Whether these small additional fluxes need to be considered will depend on the nature of the investigation.

**1 Introduction**

Evaporation is usually described as mass transfer down a concentration gradient (i.e., Fick's Law) (Monteith and Unsworth, 2008). However, strictly speaking, Fick's law is only true under isothermal conditions (Bejan, 2016, p. 639). In many hydrologic applications the surface and adjacent air temperatures are substantially different and the near-surface evaporative environment is generally not isothermal. The temperature gradient will contribute to the mass transfer via what is known as

the Soret effect that acts independently of the concentration gradient (Bejan, 2016, p. 639). The same Onsager-based framework leads to an analogous conclusion that a concentration gradient will contribute to the sensible heat transfer and this is known as the Dufour effect (Bejan, 2016, p. 639) which emphasises the symmetrical nature of the Onsager-based flux-coupling. In applications (e.g., Hydrology, Agriculture, Ecology, Climate, etc.) it is a near universal practice to ignore the Onsager-based flux-coupling. Instead we have traditionally assumed that sensible heat transfer only depends on the

temperature gradient and evaporation only depends on the concentration gradient (e.g., Monteith and Unsworth, 2008). Recent work by plant scientists suggests that *T* gradients can make an important (e.g., up to 10% of the total water vapor flux under typical circumstances) contribution to evaporation from leaves (Griffani et al., 2024). This result raises an important question of whether an approach that assumes open water evaporation to be solely described by Fick's law is sufficiently

accurate for applications in hydrology, agriculture, ecology and more generally for weather/climate studies. The aim of this

technical note is to address that question.

Ludwig (in 1856) first made a brief report noting the formation of a concentration gradient in the presence of a steady state temperature gradient in a liquid mixture. This initial observation was subsequently investigated in more detail using liquid solutions by Soret (in 1879). In liquids the phenomenon is now usually known as the Soret effect (or sometimes the Ludwig-

Soret effect). In gas mixtures, the same thermodynamic phenomenon has often been called thermodiffusion and was predicted theoretically using the kinetic theory of gases (Chapman, 1916; Enskog, 1911) before the first experimental confirmation (Chapman and Dootson, 1917). In essence, in the presence of a steady state temperature gradient there is a preferential sorting of the molecules with lighter molecules "diffusing" towards the hotter end of the gradient in the gas mixture. In air, which can be thought of as a mixture of dry air (equivalent molecular mass ≈ 29 g mol$^{-1}$) and water vapor

(molecular mass ≈ 18 g mol$^{-1}$), the (lighter) water vapor molecules will tend to "diffuse" from colder to hotter regions. Hence when an evaporating surface is colder than the adjacent air we anticipate an additional mass flux due to the temperature gradient. (Similarly, when the evaporating surface is hotter than the adjacent air we anticipate a reduction in the mass flux via the same phenomenon.) For the sake of brevity we refer to this thermal phenomenon as the Soret effect.

In the first half of the 20[th] century there was extensive interest in the Soret effect in gases because a comparison of the experimental results with theoretical calculations was routinely used to investigate the nature of the molecular collisions. A comprehensive foundation text on the topic is available (Grew and Ibbs, 1952). In this paper we make use of experimental measurements of the Soret effect in gas mixtures to evaluate the magnitude of this thermal effect on open water evaporation relative to the traditional concentration-based ('Fickian') approach.


| **Variable** | **Units** | **Description** |
|---|---|---|
| $J$ | mol m$^{-2}$ s$^{-1}$ | Total Diffusion Flux |
| $J_F$ | mol m$^{-2}$ s$^{-1}$ | Diffusion due to a concentration gradient (Fick) |
| $J_S$ | mol m$^{-2}$ s$^{-1}$ | Diffusion due to a temperature gradient (Soret) |
| 55  $c$ | mol m$^{-3}$ | Molar density of the mixture |
| $D$ | m$^2$ s$^{-1}$ | Ordinary diffusion coefficient |
| $D_T$ | m$^2$ s$^{-1}$ | Coefficient of thermal diffusion |
| $x$ | mol mol$^{-1}$ | Mole fraction of target species (e.g., water vapor) in the mixture |
| $T$ | K | Temperature of the mixture |
| 60  $k_T$ | (-) | Thermal diffusion ratio (dimensionless) |

| $\alpha_T$ | (-) | Thermal diffusion factor (dimensionless) |
| $E$ | mol m$^{-2}$ s$^{-1}$ | Evaporation from water body |

Table 1  List of key variables

## 2 Theory

Traditional thermodiffusion experiments use a binary gas mixture in a closed system where the total diffusive flux $J$ (mol m$^{-2}$ s$^{-1}$) of the target species is the sum of two terms. The first term is due to a concentration gradient (denoted by $J_F$ to represent Fick's Law) and the second due to a temperature gradient (denoted by $J_S$ to represent the Soret effect),

$$J = J_F + J_S \qquad , \qquad (1a)$$

and is formally given by (Grew and Ibbs, 1952),

$$J = \underbrace{[-c\,D\,\nabla x]}_{Fick's\ Law} - \underbrace{[c\,D_T\,\nabla(\ln T)]}_{Soret\ effect} \qquad , \qquad (1b)$$

with $c$ (mol m$^{-3}$) the molar density of the mixture, $D$ (m$^2$ s$^{-1}$) the ordinary ('Fickian') diffusion coefficient, $x$ (mol mol$^{-1}$) the mole fraction of the species of interest. $D_T$ (m$^2$ s$^{-1}$) is known as the coefficient of thermal diffusion with $T$ (K) the mixture temperature. Note that on this formulation the molar- and thermal-based diffusion coefficients have the same (classical) units for diffusivity of m$^2$ s$^{-1}$. We further note that in the recent literature it is common to define the "driving force" of

thermodiffusion using $\nabla T$ (e.g., Ortiz de Zárate, 2019; Platten, 2006; Rahman and Saghir, 2014) instead of $\nabla(\ln T)$ as used in the original Enskog-Chapman formulation based on the kinetic theory of gases. As a consequence we note that in the above-cited modern literature the thermodiffusion coefficient has a different physical meaning (and different units) from that in the original theory. Here we follow the original literature as per Eq. 1 (Chapman and Cowling, 1939; Grew and Ibbs, 1952) because nearly all of the key experimental work was completed before 1960 and was based on the classical formulation that

uses $\nabla(\ln T)$ as the "driving force". On this approach we can make direct use of extensive (translated Russian) thermodynamic tables based on pre-1960 experiments (Vargaftik, 1983) as well as many useful graphical and tabulated summaries in the foundation textbook on the topic (Grew and Ibbs, 1952).

Thermodiffusion depends on the bulk composition of the mixture. For example, the obvious limiting conditions are that $D_T$

must equal zero when $x$ equals either zero or one since there can be no identifiable thermodiffusion in a pure substance. The classical approach is to define a dimensionless thermal diffusion ratio $k_T$ (Grew and Ibbs, 1952),

$$k_T = \frac{D_T}{D} \qquad . \qquad (2)$$

The dependence on bulk composition is incorporated by expressing $k_T$ as (Grew and Ibbs, 1952),

$$k_T = \alpha_T\,x\,(1-x) \qquad , \qquad (3)$$

with $\alpha_T$ the dimensionless thermal diffusion factor. The (quadratic in $x$) form of Eq. (3) is the simplest that captures the requisite limiting conditions (i.e., $k_T = 0$ when $x = 0$ or 1). Combining Eq. 1-3 we have,

$$J = \underbrace{[-c\, D\, \nabla x]}_{Fick's\ Law} - \underbrace{[c\, D\, \alpha_T\, x\, (1-x)\frac{\nabla T}{T}]}_{Soret\ effect} \qquad . \qquad (4)$$

### 3 Empirical estimate of the thermal diffusion factor $\alpha_T$ for a $H_2O$-dry air mixture

The most extensive database on thermodiffusion in gases that we have been able to locate includes some 12 pages of summarised experimental data in a set of (translated Russian) thermodynamic tables (Vargaftik, 1983, p. 654-665) that document experimental estimates of $k_T$ and $x$ for numerous binary gas mixtures. In addition there are useful experimental data ($\alpha_T$) for a subset of the same binary gas mixtures in the foundation textbook on the topic (Grew and Ibbs, 1952). Surprisingly neither of these extensive data sources list a single experiment involving water vapor and we have not been able

to locate an experiment involving water vapor elsewhere in the scientific literature. Instead, as described below, we initially use experimental data for gas mixtures that have very similar macroscopic properties to infer the appropriate value of $\alpha_T$ for the $H_2O$-dry air mixture of primary interest.

Foundation work by Chapman established that the molecular masses of the mixture components only influence the thermal

diffusion factor by their ratios (Chapman, 1940). The two macroscopic (dimensionless) variables traditionally used to collate the various theoretical/numerical results are known as the proportionate mass difference $M$ ($= (m_1 - m_2) / (m_1 + m_2)$, with $m_1$ the molecular mass of the heavier component) and the proportionate diameter difference $\Sigma$ ($= (\sigma_1 - \sigma_2) / (\sigma_1 + \sigma_2)$, with $\sigma_1$ the diameter of the heavier component) defined by the collision diameters ($\sigma$) of the mixture components (Chapman, 1940; Grew and Ibbs, 1952). In general, the thermal diffusion factor $\alpha_T$ increases with $M$ and $\Sigma$, but is much more sensitive to $M$ than to

$\Sigma$ (Grew and Ibbs, 1952, Fig. 7, p. 29). The values of $M$, $\Sigma$ for three binary gas mixtures ($N_2$-$CO_2$, $N_2$-$N_2O$, $H_2$-$CO_2$) for which we have extensive experimental data (Vargaftik, 1983) are listed in Table 2 along with relevant values for a $H_2O$-dry air mixture for which we have no experimental data. We note that the values of $M$, $\Sigma$ for the $H_2O$-dry air mixture are more or less the same as those for the two $N_2$-based mixtures.



| Gas Mixture | $M$ | $\Sigma$ |
|---|---|---|
| $N_2$-$CO_2$ | 0.22 | 0.08 |
| $N_2$-$N_2O$ | 0.22 | 0.08 |
| $H_2$-$CO_2$ | 0.91 | 0.46 |
| $H_2O$-dry air | 0.23 | 0.07 |

Table 2: Dimensionless variables ($M$, $\Sigma$; see main text) for four binary gas mixtures. Calculations for $\Sigma$ use the following values for the collision diameter: $\sigma(N_2) = 3.7 \times 10^{-10}$ m, $\sigma(CO_2) = 4.3 \times 10^{-10}$ m, $\sigma(N_2O) = 4.3 \times 10^{-10}$ m, $\sigma(H_2) = 1.6 \times 10^{-10}$ m, $\sigma(\text{dry air}) = 3.7 \times 10^{-10}$ m that have been calculated using an empirical formula dependent on the molecular mass (Wang and Frenklach, 1994, see their Eq. 6).


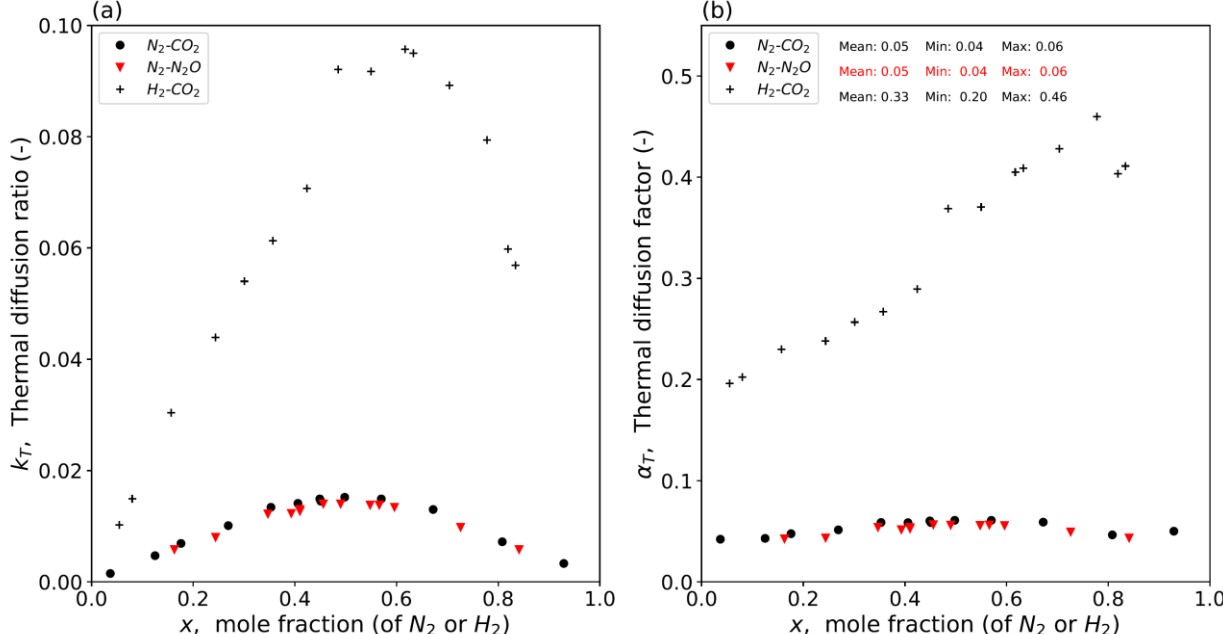

Figure 1: Experimental data for three binary gas mixtures at a mean $T$ of 328 K (Vargaftik, 1983, p. 661). Composition dependence of the (a) thermal diffusion ratio ($k_T$) and (b) the thermal diffusion factor ($\alpha_T$, calculated per Eq. 3).

The experimental data for the three binary gas mixtures are shown in Fig. 1. The experimental results highlight the strong dependence of $k_T$ on the bulk mixture composition (Fig. 1a). With data for $k_T$ and the mole fraction (of $N_2$ and $H_2$) available in the tables we calculate the dimensionless thermal diffusion factor $\alpha_T$ for all three gas mixtures (Fig. 1b). The results show that the experimental values of $\alpha_T$ for the two $N_2$-based gas mixtures are more or less the same (Mean $\approx 0.05$) (Fig. 1b). The experimental values of $\alpha_T$ summarised by Grew and Ibbs (1952, p. 130) for the $N_2$-based mixtures are consistent with those

given here (Fig. 1b). In contrast, the experimental values of $\alpha_T$ for the $H_2$-$CO_2$ mixture are much larger as expected based on

the much larger value for $M$ (Grew and Ibbs, 1952, see their Fig. 8 on p. 30). We know by experiment that $\alpha_T$ usually increases slightly with $T$ and our empirical estimates of $\alpha_T$ are for a $T$ of 328 K (Fig. 1). We can use the empirical equation of Youssef et al. (1965, their Eq. 7) for the $N_2$-$CO_2$ mixture to correct the value of $\alpha_T$ back to a $T$ of 300 K and doing that we find a value of 0.048. In the context of the magnitude of the Soret effect relative to concentration-dependent diffusion we show below that it is not necessary to consider the slight $T$ dependence because the Soret effect is already relatively small. This result suggests a value of 0.05 for the thermal diffusion factor would be a useful starting point for the $H_2O$-dry air mixture.

The thermodynamic tables used above (Vargaftik, 1983) list $k_T$ as a function of $x$ at a given $T$. An alternative approach to estimate $\alpha_T$ is to use an empirical correlation based on the proportionate mass difference $M$. For that we used a tabular summary of $\alpha_T$ based on 113 individual thermodiffusion experiments involving 39 different binary gas mixtures (Grew and Ibbs, 1952, their Table VA on p. 128-130). The resulting empirical equation was $\alpha_T \approx 0.38\,M$ ($R^2 = 0.92$, n =113, see Fig. A1 in Appendix A) and confirmed the strong dependence on $M$ noted in earlier works (Chapman, 1940). With that equation we predict $\alpha_T \approx 0.09$ for the $H_2O$-dry air mixture. This is a nearly a factor of two higher than our earlier estimate ($\alpha_T \approx 0.05$) derived from individual experiments using two $N_2$-based gas mixtures. Here we follow a conservative approach and adopt the larger value as suitable for the $H_2O$-dry air mixture.

We use that value (i.e., $\alpha_T = 0.09$) in the next section to evaluate the magnitude of the Soret effect relative to the effect of concentration-dependent diffusion using a simple example.

## 4 Typical magnitude of the Soret effect relative to ordinary 'Fickian' diffusion

We begin by rewriting Eq. 4 with the adopted value for $\alpha_T$ (= 0.09) for the $H_2O$-dry air mixture and we follow the typical hydrologic sign convention (diffusion is positive upwards from the liquid surface) with the total diffusive flux of water vapor given by,

$$J = \left[\frac{c\,D}{\Delta z}\,(x_s(T_s) - x_a)\right] - \left[\frac{c\,D}{\Delta z}\left(0.09\,\bar{x}\,(1-\bar{x})\,\frac{(T_s - T_a)}{\bar{T}}\right)\right] \quad , \quad (5)$$

with $x_s(T_s)$ the (saturated) mole fraction of water vapor at the evaporating surface of temperature $T_s$, $x_a$ the mole fraction of water vapor in air having temperature $T_a$, $\bar{x}$ is the mean mole fraction (= $(x_s(T_x) + x_a)/2$) of water vapor over the diffusive pathway of thickness $\Delta z$ and $\bar{T}$ (= $(T_s + T_a)/2$) the mean $T$ over the same pathway. Note that we will have a positive Soret effect when $T_s < T_a$. In writing Eq. 5 we follow the standard threshold model for the boundary layer with the concentration and temperature assumed to both follow a linear profile from the surface to the top of the boundary layer a distance $\Delta z$ (~ 1-5 mm depending primarily on wind speed) above the liquid surface (Lim et al., 2012; their Fig. 1).

To take a typical numerical example, assume air at 298 K and relative humidity of 60% with total air pressure of 1 bar where the surface $T$ is initially 5 K cooler than the air. With those data (Table 3) we have,

$$J = \frac{c\,D}{\Delta z}\left[(0.0232 - 0.0189) - ((0.09)(0.0211)(0.9789)\left(\frac{293-298}{295.5}\right)\right] \qquad , \qquad (6a)$$

which equals,

$$J = \frac{c\,D}{\Delta z}[0.0043 + 0.000031] \qquad . \qquad (6b)$$

In this numerical example, concentration-dependent mass diffusion contributes 99.3% of the total diffusion flux with the Soret effect contributing 0.7% to the total. We could enhance the percentage due to the Soret effect by holding the air properties constant while decreasing the surface $T$ until, for example, $x_s(T_s)$ equals $x_a$. In that limit the mole fraction gradient would become zero and Soret effect would then be 100% of the total flux. However, the flux due to the Soret effect remains small as we show below.

| Variable | Value | Units |
|---|---|---|
| $x_s(T_s)$ | 0.0232 | mol mol$^{-1}$ |
| $x_a$ | 0.0189 | mol mol$^{-1}$ |
| $\bar{x}$ | 0.0211 | mol mol$^{-1}$ |
| $T_s$ | 293 | K |
| $T_a$ | 298 | K |
| $\bar{T}$ | 295.5 | K |

**Table 3 Data for numerical example: standard air at 298 K and rel. humidity = 60% with $T_s$ = 293 K**

Having developed an initial sense for the typical magnitude of the Soret effect we use experimental data in the next section to evaluate the relative Soret effect over a much larger range of earth surface conditions.

**5 Contribution of the Soret effect to the total evaporation**

To make a more comprehensive assessment of the magnitude of the Soret effect we use a recently published experimental database on evaporation (Roderick et al., 2023). The database includes 70 individual evaporation experiments made under carefully controlled laboratory conditions with $E$ measured using an accurate balance. The experiments encompass a broad range of environmental conditions ($T_a$ range from 15 to 45°C, relative humidity range from 20 to 78%, wind speed ($U$) range from 0.5 to 4 m s$^{-1}$). One important feature of the experiments is that the air properties ($T$, humidity, wind) were held constant during each individual evaporation experiment which meant that the water bath from which evaporation occurred

was a very good approximation to the classical theoretical wet bulb thermometer. As a consequence the water bath was generally colder than the adjacent air which would tend to maximise the (positive) contribution of the Soret effect and these data are ideal for assessing the relative importance of the Soret effect under typical conditions at the earth's surface.

Traditionally, hydrologists have assumed that the total evaporative flux from the surface to the air was due to diffusion
alone, but the broader engineering literature has long formulated the problem slightly differently. In particular while water vapor is diffusing along a (concentration and/or temperature) gradient from the surface to the air there must be a counter diffusion of dry air in the opposite direction. The liquid surface is highly permeable to water but to a reasonable approximation (i.e., ignore Henry's Law solubility) it is nearly impenetrable to the dry air. With all else equal, the dry air should accumulate at the liquid-air interface via the diffusion flux but that is not observed. Instead, by an argument originally
due to Stefan there must be a compensating upwards flow that exactly balances the downward diffusion of dry air leaving the overall velocity of the dry air in the boundary layer to be near zero. This compensating advective flux is widely recognised in the broader engineering literature (e.g., Kreith et al., 1999; Cussler, 2009) but has generally been ignored in hydrology and other closely related climate-based fields (Kowalski, 2017). Here we incorporate this additional advective contribution ($E_A$) and express the total surface-air evaporative mass flux ($E$) as,

$$E = J_F + J_S + E_A \qquad , \qquad (7)$$

with the additional advective flux defined by (Kreith et al., 1999; Cussler, 2009),

$$E_A = x_a E \qquad . \qquad (8)$$

The derivation of Eq. 7-8 is fully described in Appendix B. Hence, the total evaporative flux is given by,

$$E = \underbrace{\left[\frac{cD}{\Delta z}\left(x_s(T_s) - x_a\right)\right]}_{Fick's\,Law} - \underbrace{\left[\frac{cD}{\Delta z}\left(0.09\,\bar{x}\,(1-\bar{x})\,\frac{(T_s - T_a)}{\bar{T}}\right)\right]}_{Soret\,effect} + \underbrace{x_a E}_{Advection} \qquad . \qquad (9)$$


To split the observed evaporation into the separate components we use Eq. 9 to first calculate the value of $\Delta z$ that gave the observed evaporation with the concentration-dependent diffusion coefficient for water vapor in air calculated at the mean temperature. The (calculated) values of the boundary layer thickness (see Fig. C1a in Appendix C) are typically in the range 1-5 mm and decline with wind speed as expected (Lim et al., 2012). Further, in all but one of the 70 experiments, the
evaporating liquid water surface was colder than the adjacent air; by up to ~ 17°C in a few instances (Fig. C1b in Appendix C). Typically, the evaporating surface was ~ 10°C colder than the adjacent air and we reiterate that these conditions will maximise the (positive) contribution of the Soret effect to the overall evaporative flux. The results of the flux partitioning are summarised in Fig. 2.

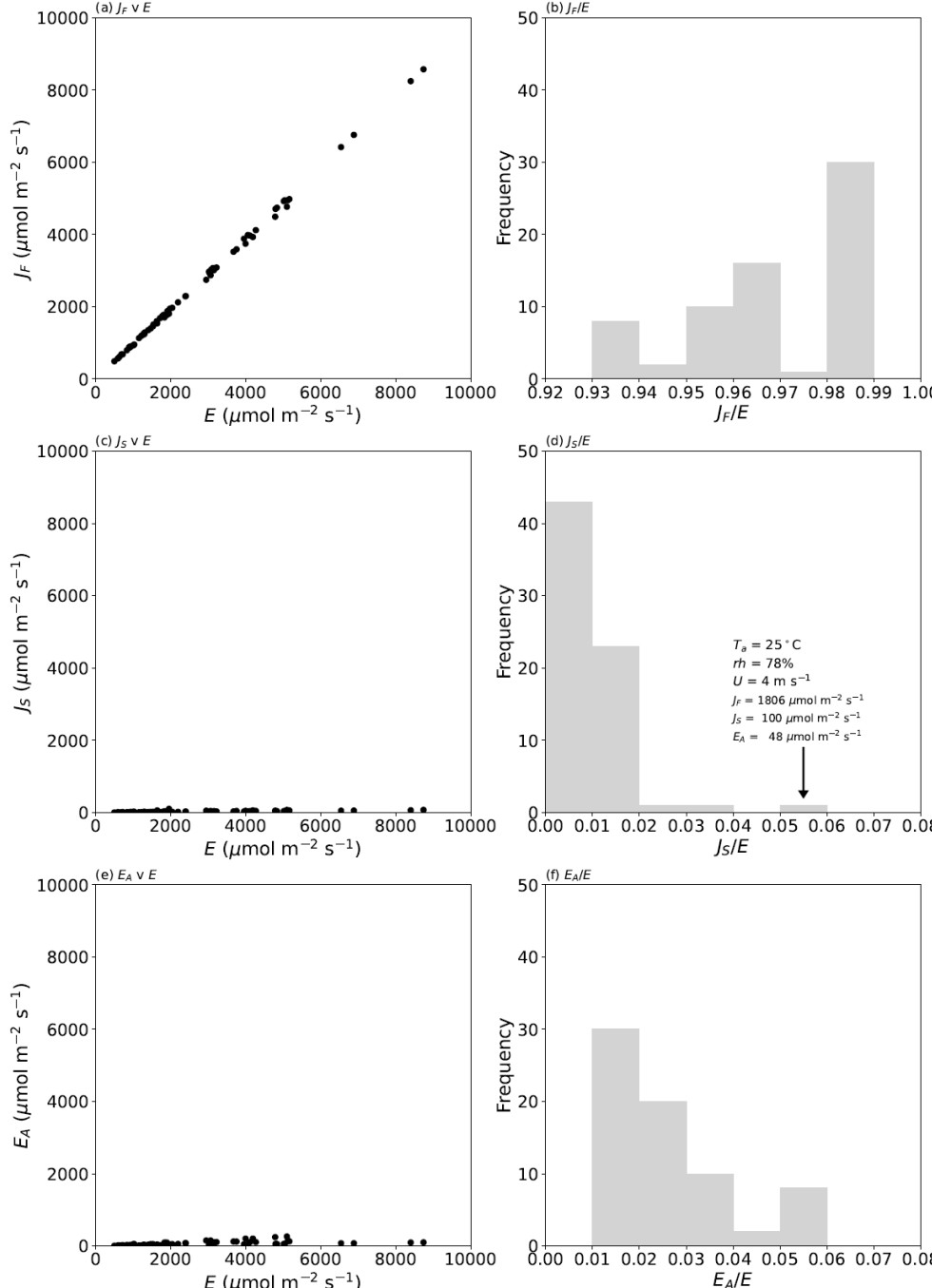

**Figure 2 Partitioning the measured evaporative flux in 70 laboratory experiments into separate diffusive and advective components. The first column shows the evaporation due to the (a) concentration gradient ($J_F$), (c) temperature gradient ($J_S$) and the (e) advective flow ($E_A$). The right column (bdf) depicts histograms of the relative fractions of each component. The text on panel (d) denotes the air properties and fluxes during that experiment. See Fig. D1 (Appendix D) for more detail on (c) and (e).**

The results confirm the total evaporative flux to be dominated by Fickian diffusion ($> 93\%$ of total in all cases, Fig. 2b). The next most important term is the advective component which scales directly with the mole fraction of water in the air (Eq. 8). In our experiments the advective component was responsible for as much as 5% of the total flux (Fig. 2f) in one extreme set of laboratory experiments ($T_a$ at 45°C and relative humidity $\sim 52\%$) and such conditions occasionally occur in the current climate (e.g., Schär, 2016). More generally the advective component would regularly represent up to $\sim 4\%$ of the total

evaporative flux in moist tropical regions in the current climate but will be smaller in colder climates (Shakespeare and Roderick, 2024). We conclude that the experimental data for the advective flux (Fig. 2f) are indicative of the range of real world conditions in the current climate.

The impact of the Soret effect was typically smaller again and was less than 2% of the total evaporative flux in most (67 of

70) experiments (Fig. 2d). The maximum relative Soret effect was 5% in a sole experiment (see text label on Fig. 2d). That experiment was conducted under very high relative humidity (78%, Fig. 2d) which reduced the total evaporation (to 1954 µmol m$^{-2}$ s$^{-1}$, see Fig. 2d). The total evaporation in that instance is actually a small flux (cf. Fig. 2a) and this circumstance will always enhance the relative Soret effect. To see that more clearly we note that the relative Soret effect ($= J_S/E$) can be closely approximated by the ratio $J_S/J_F$. Using Eq. 7 and 9 we have for that ratio,

$$\frac{J_S}{E} \approx \frac{J_S}{J_F} = -\alpha_T \, \bar{x} \, (1 - \bar{x}) \, \frac{(T_s - T_a)}{\bar{T}} \, \frac{1}{(x_s(T_s) - x_a)} \qquad . \qquad (10)$$

For typical earth surface conditions we have $(1 - \bar{x}) \to 1$. Using that approximation and replacing the mean values by the relevant sums we have a more convenient expression as follows,

$$\frac{J_S}{E} \approx \frac{J_S}{J_F} \approx -\alpha_T \left( \frac{x_s(T_s) + x_a}{x_s(T_s) - x_a} \right) \left( \frac{T_s - T_a}{T_s + T_a} \right) \qquad . \qquad (11)$$

With this form we note that the unfamiliar relative Soret effect will actually be familiar to many readers from hydrology as

well as the broader earth science communities as nothing more than a rescaled Bowen ratio ( $\beta \propto [T_s - T_a]/[x_s(T_s) - x_a]$ ). By definition the relative Soret effect increases with the surface to air $T$ difference. By inspection of Eq. 11 it is also clear that the relative Soret effect will tend to be higher as the (i) $T$ increases since the sum $(x_s(T_s) + x_a)$ also tends to increase in warmer climates, and, as the (ii) surface to air mole fraction difference tends to zero, i.e., as $(x_s(T_s) - x_a) \to 0$.


To test whether our experimental data was indicative of the entire range of possibilities we conducted a broad literature survey to identify extreme circumstances under which we might expect a high relative and absolute Soret effect. To that end we identified the 2015 Persian Gulf heatwave (Schär, 2016) as an event with the key attributes (warm water, large surface to air $T$ difference). At midday on 31 July 2015 the observed conditions ($T_a$ = 45°C, relative humidity $\sim 45\%$, $T_s$ = 32°C) were

the most extreme (Schär, 2016) and we estimate that the Soret effect would have accounted for about 5% of the total

evaporative flux at this time (Table E1, Appendix E). The magnitude of the fluxes would vary primarily with wind speed (i.e., with the boundary layer thickness, $\Delta z$) and assuming windy conditions we estimate the Soret flux was around 184 µmol m$^{-2}$ s$^{-1}$ (latent heat flux equivalent = 8.0 W m$^{-2}$) while the total evaporative flux was 4085 µmol m$^{-2}$ s$^{-1}$ (176.7 W m$^{-2}$) (Table E1, Appendix E). Despite the fact that the relative Soret effect was ~ 5%, the absolute value of the Soret flux (8 W m$^{-2}$ latent heat flux equivalent) is actually relatively small in a hydroclimatic context. Further, heatwave conditions are often accompanied by calm conditions and if that was the case the Soret flux would be much smaller (~ 37 µmol m$^{-2}$ s$^{-1}$, 1.6 W m$^{-2}$ latent heat flux equivalent) with the total evaporative flux declining to ~ 817 µmol m$^{-2}$ s$^{-1}$ (35.4 W m$^{-2}$) (Table E1, Appendix E). This extreme example emphasises the point that the relative Soret effect will usually increase as the total flux declines.

These data beg the question of whether the flux due to the Soret effect would be routinely measureable? In our experiments the evaporation was measured using an accurate balance under carefully controlled laboratory conditions which enabled us to also estimate the error in each steady-state evaporation measurement. The maximum evaporation error in any single evaporation experiment was ~ 60 µmol m$^{-2}$ s$^{-1}$ (latent heat flux equivalent is 2.6 W m$^{-2}$) (Fig. 3). In contrast the flux due to the Soret effect was in general slightly larger than the measurement error implying that it was resolvable in most of the 70 carefully controlled laboratory experiments (Fig. 3). The maximum flux due to the Soret effect reported here is ~ 100 µmol m$^{-2}$ s$^{-1}$ (latent heat flux equivalent is 4.3 W m$^{-2}$) for the same experiment discussed previously. Whether fluxes of that magnitude are important would depend on the circumstances but they are unlikely to be a major problem given a typical measurement resolution for the latent heat flux of 10-20 W m$^{-2}$ in most field-based programs.

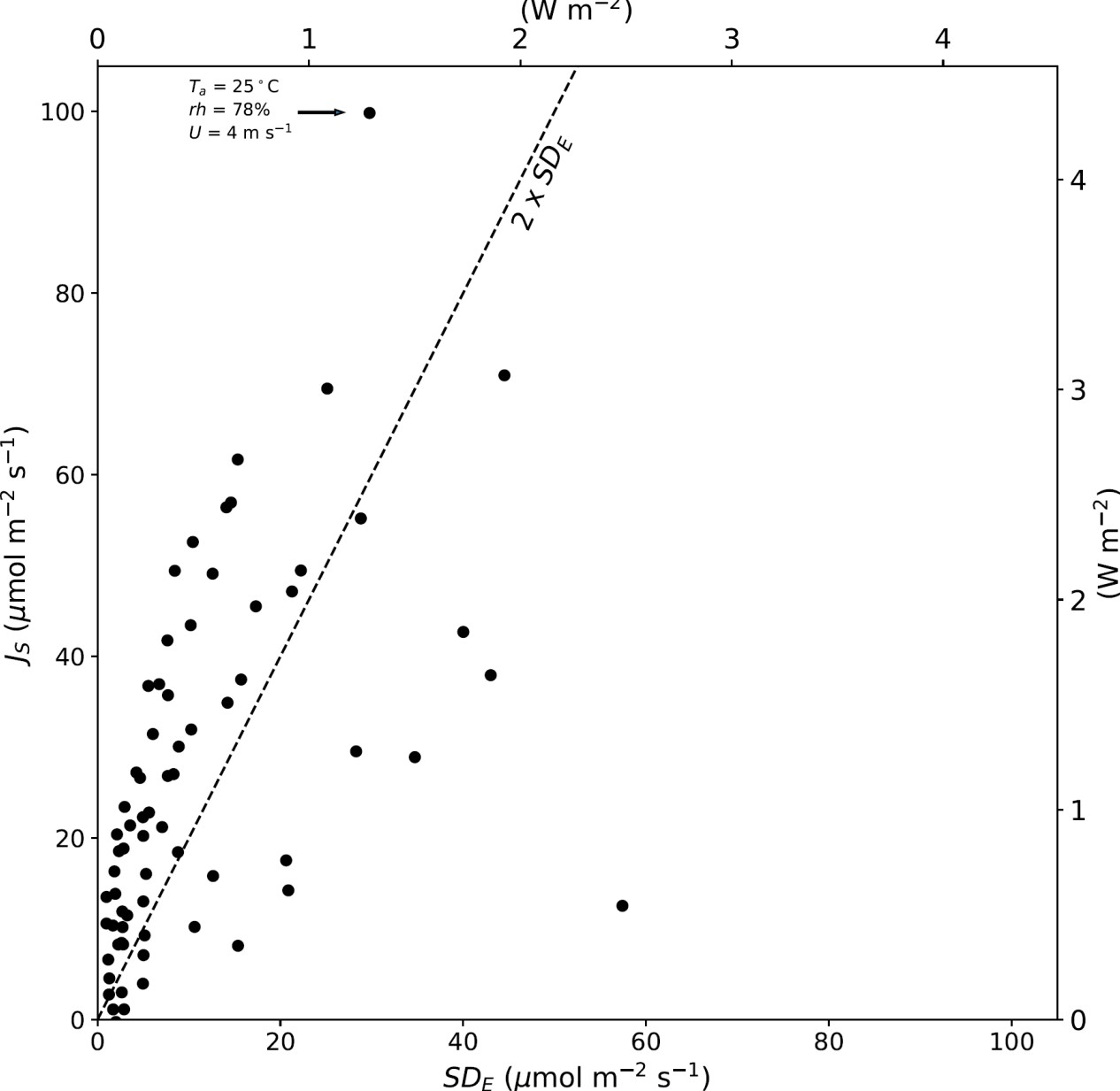

Figure 3 Magnitude of the Soret effect ($J_S$) compared to the standard deviation in the total evaporative flux ($SD_E$) in 70 individual laboratory-controlled evaporation experiments. The dashed line denotes the 90% confidence interval ($2 \times SD_E$). The text near the uppermost $J_S$ (~ 100 μmol m$^{-2}$ s$^{-1}$) summarise the air properties for that particular evaporation experiment. The latent heat flux equivalent (secondary x and y axes) assumed the latent heat of vaporisation was 2.4 MJ kg$^{-1}$.

## 5 Discussion and Conclusions

The Soret effect is a real phenomenon and played a prominent role in the overall development of the kinetic theory of gases (Chapman and Cowling, 1939; Grew and Ibbs, 1952). However, we were unable to locate a single experiment involving water vapor and instead we initially assumed that the thermal diffusion factor for a $H_2O$-dry air mixture was the same as for other well studied binary gas mixtures ($N_2$-$CO_2$, $N_2$-$N_2O$) (Fig. 1) having very similar relative differences in molecular mass between the components (Table 2). That experimentally-based approach led to a numerical value of 0.05 for the thermal diffusion factor in a $H_2O$-dry air mixture. We then augmented that by an analysis using data from 113 individual experiments (all completed before 1952) based on 39 different binary mixtures of gases and the resulting correlation (Fig. A1) was used to estimate the thermal diffusion factor to be 0.09 for the same $H_2O$-dry air mixture. We followed a conservative approach by adopting the larger value ($\alpha_T \sim 0.09$) in subsequent analysis but we recognise that in the absence of a specific $H_2O$-dry air experiment these values must be considered as interim estimates.

Recent work has also examined the impact of the Soret effect on evaporation (i.e., transpiration from plants) but used a numerical value of 0.5 for the (same) thermal diffusion factor (Griffani et al., 2024). This is a factor of five larger than the value we adopted and they predicted that the magnitude of the Soret effect was also typically larger by a factor of five. This factor of five difference requires an explanation.

The Griffani et al. (2024) result is different for two reasons. First, it was based on Landau's original theoretical derivation which assumed a binary gas mixture having perfect elastic collisions where the molecular mass of the heavier molecule was assumed to be substantially larger than that of the lighter molecule (Lifshitz and Pitaevskii, 1981, p. 36). The latter assumption implies a value for the proportionate mass difference $M$ of 1 which is much larger than the actual value for a $H_2O$-dry air mixture ($M = 0.23$, Table 2). Further, the measured thermal diffusion factor in real gas mixtures is substantially less (by ~ 20-70%, see Table VA in Grew and Ibbs (1952, p. 128-130)) than the theoretical calculation based on the perfectly elastic collision assumption. Hence because of the underlying assumptions the Griffani et al. (2024) estimate for the thermal diffusion factor is not relevant to a $H_2O$-dry air mixture.

When the surface is colder than the air the Soret effect predicts an additional evaporative flux and the reverse holds when the surface is warmer than the air. We used an existing experimental data to evaluate the magnitude of the Soret effect. In the experimental data the evaporating surface was colder than the air and our experimental assessment (Fig. 2) was restricted to a positive Soret effect. Our results using a thermal diffusion factor of 0.09 for the $H_2O$-dry air mixture show that the Soret effect typically enhances the evaporative flux by up to 2% (Fig. 2d) and this effect was small in an absolute sense (usually less than 60 μmol m$^{-2}$ s$^{-1}$, < 2.6 W m$^{-2}$ latent heat flux equivalent) (Fig. 3). Using the most extreme example we could identify (2015 Persian Gulf Heatwave, $T_a = 45°C$, relative humidity ~ 45%, $T_s = 32°C$) we found the relative Soret effect in

that instance was also ~ 5% of the total evaporative flux but again the absolute effect remained small (< 184 µmol m$^{-2}$ s$^{-1}$, < 8 W m$^{-2}$ latent heat flux equivalent) (Table E1, Appendix E). In summary, when a liquid water surface is cooled relative to the air, the relative Soret effect increases but the total evaporative flux is reduced. Hence it is important to also consider the absolute magnitude of the Soret flux. The alternate conditions are when the surface is warmer than the air. These conditions would lead to a negative Soret effect thereby reducing the total evaporative flux but the relative impact is likely to remain small under most conceivable conditions since those same conditions will increase the total evaporative flux. To give an example, assume the same conditions as in Table 2 ($T_a$ = 298 K, relative humidity = 60%) but we now increase the surface temperature to say 372 K (i.e., just below the boiling point). The result is that the Soret effect reduces the total flux by only 0.5% and can be safely ignored. In summary we anticipate the Soret effect to remain small in most typical circumstances (usually less than 60 µmol m$^{-2}$ s$^{-1}$, < 2.6 W m$^{-2}$ latent heat flux equivalent) (Fig. 3) but may become a factor of 2 or 3 larger under exceptional circumstances when the surface is much cooler than the air under very warm conditions. Whether those magnitudes are important will depend on the application but for most applications in hydrology they can likely be ignored as has been common practice for the last century.

We also assessed the importance of the (so-called Stefan) advective flux and note that this always enhances the diffusive flux in an apparently straight-forward way (Kowalski, 2017) and is much simpler to implement compared to the Soret effect. On a molar basis this flux scales directly with the overall mole fraction of water vapor and can be as large as 5% of the total flux (Fig. 2f). Again, whether that is important will depend on the application but there is no obvious reason to ignore it since it can be readily calculated (Eq. 8).

Under typical environmental conditions the Soret effect is also small in liquids. Despite that, there is current interest in evaluating whether the Soret effect can be used to desalinate water (Xu et al., 2024). That approach requires innovative engineering by recycling the treated stream multiple times to eventually separate the salt from the fresh water (Xu et al., 2024). Hence despite the fact that the Soret effect is small under typical environmental conditions it may still have important future engineering applications.

**Author Contribution**

MLR conceived the overall project. MLR and CJS undertook the analysis. MLR prepared the manuscript with contributions from CJS.

**Data Availability**

The evaporation experiment data is available at https://doi.org/10.5281/zenodo.8381685.

## Competing Interests

The authors declare that they have no competing interests.

## Acknowledgements

We thank Pierre Rognon, Graham Farquhar and Fulton Rockwell for discussions on the Soret effect. We also thank five anonymous reviewers as well as Dr Koutsoyiannis for helpful comments that improved the manuscript. We acknowledge Dr Kowalski for pointing out the missing advective flow in the original analysis. The research was supported by the Australian

Research Council (DP190100791).

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

## Appendix A – Empirical dependence of $\alpha_T$ on $M$

We use tabulated experimental data (from 113 individual experiments) based on 39 different binary gas mixtures (Grew and Ibbs, 1952) to plot the thermal diffusion factor $\alpha_T$ as a function of the proportionate mass difference $M$ (Fig. A1). The variations at a given $M$ (i.e., the apparent vertical lines) are due to (the sometimes very large) experimental variations in $T$ imposed on the same binary gas mixture. The plot includes data over a total $T$ range from 89 to 755 K but restricting the analysis to a more relevant $T$ range ($260 < T < 360$) did not materially alter the regression result (See Fig. A1 caption). The cluster near the origin are experiments involving isotopic mixtures. Two outliers have been highlighted and both are from a single study that used binary gas mixtures incorporating Radon gas which had the largest molecular mass ($\approx 222$ g mol$^{-1}$) of any substance in the experimental summary.

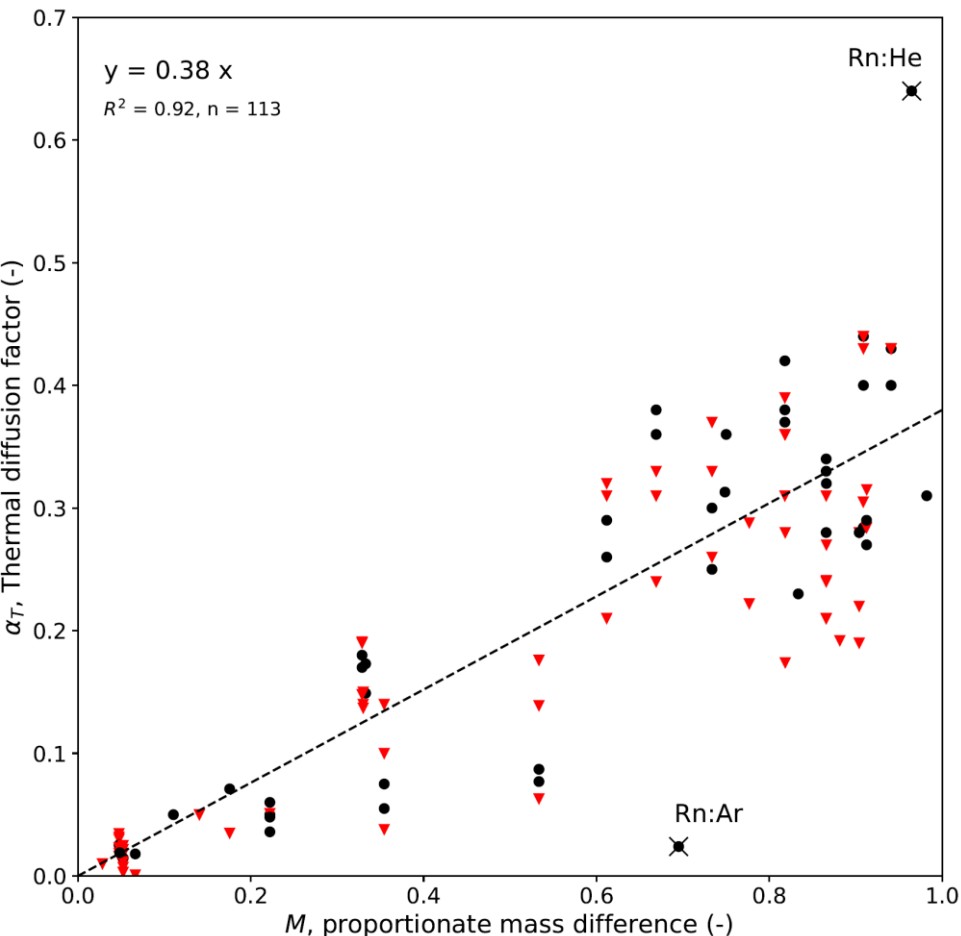

**Figure A1 Empirical dependence of $\alpha_T$ on $M$ based on experimental data from binary gas mixtures (Grew and Ibbs, 1952; their Table VA on p. 128-130). The regression results use all available data (text at top left) with the black dots denoting individual experiments with mean $T$ in the 260-360 K range (regression: y = 0.39 x, R$^2$ = 0.91, n = 47) and the remaining experiments (n = 66) shown as red inverted triangles. The two outliers (x: Rn:Ar, Rn:He) are binary gas mixtures incorporating Radon gas.**

## Appendix B – Derivation of Equations 7-9 in the main text

With reference to the main text, the (so-called Stefan) compensating flow implies that the total evaporative flux is the sum of (two) diffusive mass fluxes augmented by an advective mass flux (Kreith et al., 1999; Cussler, 2009),

$$E = J_F + J_S + E_A \qquad . \tag{B1}$$

An important point for evaporation from the earth's surface is that the total mass flux is dominated by the mass flux of water vapor from which it follows that $E$ is also a very good approximation to the total liquid-air mass flux at the surface (Kowalski, 2017). To derive an expression for $E_A$, we first note that (Kreith et al., 1999; Cussler, 2009),

$$E = c_w v_w \qquad , \tag{B2}$$

where $c_w$ (mol m$^{-3}$) is the molar concentration of water vapor in the air and $v_w$ (m s$^{-1}$) the velocity of the water vapor which is measured relative to laboratory frame of reference. To split the total flux into separate components due to diffusion and advection we specify a reference velocity ($v^*$, defined below) as follows (Kreith et al., 1999; Cussler, 2009),

$$E = c_w (v_w - v^*) + c_w v^* \qquad . \tag{B3}$$

The first term on the left denotes the flux due to diffusion and the latter is the advective component. Hence we have,

$$J_F + J_S = c_w (v_w - v^*) \qquad , \tag{B4}$$

and,

$$E_A = c_w v^* \qquad . \tag{B5}$$

Here we define the reference velocity as the molar-average velocity (also equal to the volume-average velocity assuming constant density) defined as (Kreith et al., 1999; Cussler, 2009),

$$v^* = x\, v_w + (1 - x) v_d \qquad , \tag{B6}$$

with $v_d$ (m s$^{-1}$) the velocity of the dry air. The formulation adopted here (i.e., downward diffusion of dry air is perfectly compensated by an upwards advective flow) implies that $v_d$ is zero and Eq. B6 becomes,

$$v^* = x\, v_w \qquad . \tag{B7}$$

Combining that result with Eq. B2 and B5 we have,

$$E_A = x\, E \qquad . \tag{B8}$$

Hence Eq. B1 can be rewritten as,

$$E = \left(\frac{1}{1-x}\right)(J_F + J_S) \qquad . \tag{B9}$$

In the engineering literature the prefactor ($= \frac{1}{1-x}$) in Eq. B9 is often called a Stefan pre-factor or sometimes a Stefan adjustment and this formulation is often described as the classic "diffusion through a stagnant gas film" problem (Kreith et al., 1999; Cussler, 2009). Importantly, this formulation works best with constant density and is best evaluated at the top of the boundary layer (i.e., a distance $\Delta z$ above the liquid surface) where the air density is (typically assumed) reasonably constant thereafter. In our formulation $x_a$ is the mole fraction of water at the top of the boundary layer as appears in the main text (see Eq. 8).

## Appendix C – Key characteristics of the evaporation experiments

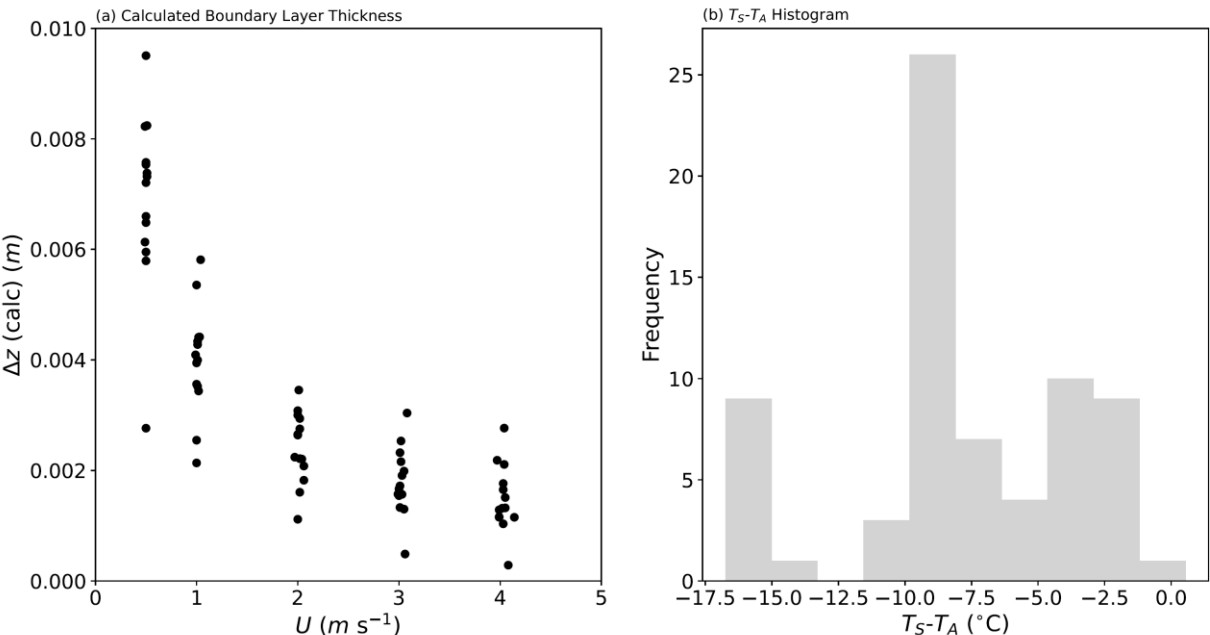

**Figure C1 Key characteristics of the 70 individual laboratory-controlled evaporation experiments. (a) Calculated value of the boundary layer thickness (see main text) as a function of windspeed ($U$) and (b) histogram showing the variation in $T_s$-$T_a$ over the 70 individual evaporation experiments.**


**Appendix D – Alternate plots for Fig. 2c and 2e**

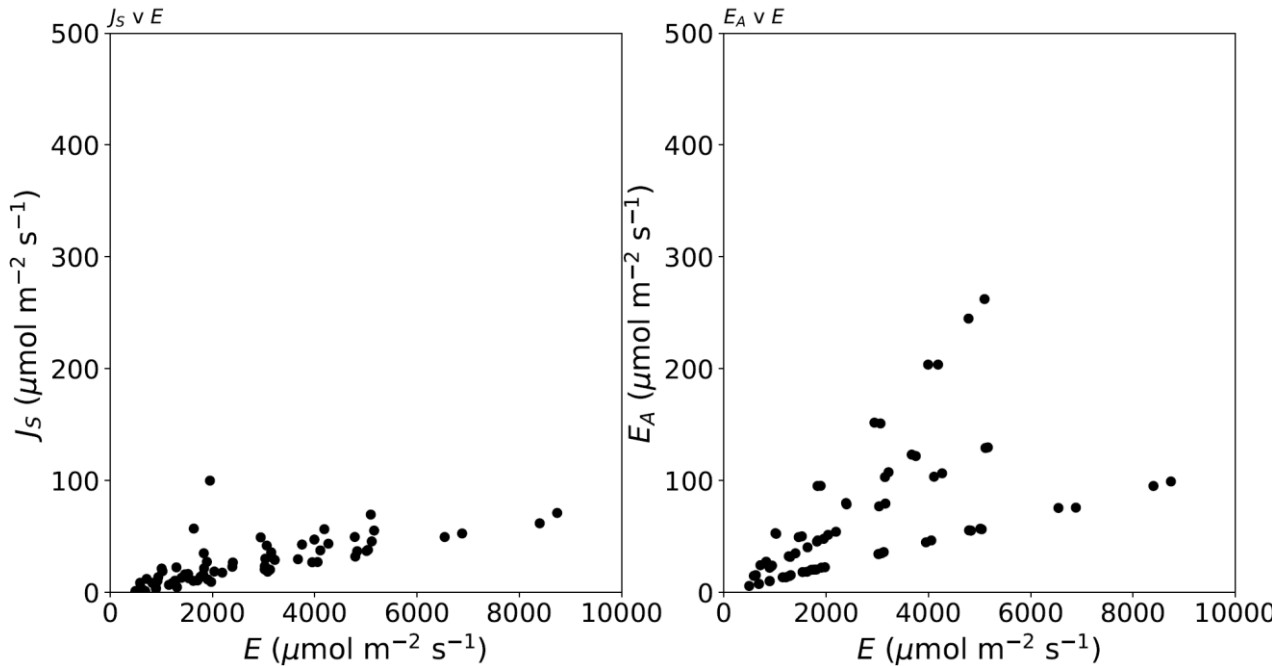

**Figure D1 Additional detail (different y axis) for (left) Fig. 2c and (right) Fig. 2e in main text**

**Appendix E – The Soret effect during the 2015 Persian Gulf heatwave**

| Heatwave in the Persian Gulf (Midday, 31 July 2015) | | |
|---|---|---|
| **DATA:** | | |
| $T_s = 305$ K, $T_a = 318$ K, $x_s(T_s) = 0.0476$ mol mol$^{-1}$, $x_a = 0.0441$ mol mol$^{-1}$ $c = 38.61$ mol m$^{-3}$, $D = 2.76 \times 10^{-5}$ m$^2$ s$^{-1}$ | | |
| **CALCULATIONS:** | | |
| $J_S/E$ | ~ 0.05 | |
| | $\Delta z = 0.005$ m **(Calm conditions)** | $\Delta z = 0.001$ m **(Windy conditions)** |
| $J_F$ (µmol m$^{-2}$ s$^{-1}$) | **746** [32.3 W m$^{-2}$] | **3730** [161.3 W m$^{-2}$] |
| $J_S$ (µmol m$^{-2}$ s$^{-1}$) | **37** [1.6 W m$^{-2}$] | **184** [8.0 W m$^{-2}$] |
| $E_A$ (µmol m$^{-2}$ s$^{-1}$) | **34** [1.5 W m$^{-2}$] | **171** [7.4 W m$^{-2}$] |
| $E$ (**total**) (µmol m$^{-2}$ s$^{-1}$) | **817** [35.4 W m$^{-2}$] | **4085** [176.7 W m$^{-2}$] |


**Table E1 Estimate of the evaporative fluxes during an extreme heatwave in the Persian Gulf. The observed data are from Schär (2016, their Fig. 1) and are taken at midday on 31 July 2015 ($T_a = 45°$C, rel. humidity = 45%, $T_s = 32°$C and assumed equal to the midpoint of the observed dewpoint and wet-bulb $T$). The wind conditions were not described in the original publication and we completed calculations assuming a typical value for the boundary layer thickness under the stated (calm, windy) conditions. We**

**use a fixed value for the latent heat of vaporisation (2.4 MJ kg$^{-1}$) to convert the evaporative flux to the latent heat flux equivalent [in W m$^{-2}$].**