# Peer review of "Technical note: An assessment of the relative contribution of the Soret effect to open water evaporation"

_EGUsphere, 2024_

## Referee Comment (RC1)

**Distinguishing transport types**

In the following examples, I will attempt to distinguish clearly between two types of transport, initially in the simplest of situations and then towards increasing complexity. The first type I will call Newtonian transport, whereby conveyance of a fluid constituent occurs because of fluid motion. The second I will call Fickian transport, whereby transport of a fluid constituent occurs independent of, or most simply in the absence of, fluid motion. I present examples of each that are quite clear, but then two cases that may be less intuitive to the authors (or other scientists who specify Fick's 1$^{st}$ law using molar concentration gradients), illustrating that Newton's laws should be kept in mind when specifying Fick's law.

**1 Newtonian transport**

The first case involves a system composed of two elements: a fluid that is pure xenon gas (Xe) and "Isaac", a cubic container, so named because he will help us to interpret the situation using Newton's laws. Initially, the system and indeed both elements are at rest, and importantly no external forces act on the system, but the Xe occupies only the right half of the container, leaving a vacuum on the left as depicted below.

[Figure]

Of course, this situation is not stable, and so the Xe will expand to occupy the whole container. However, if we ask Isaac about the motion of the system and its elements, he would say the following: "Newton's three laws explain the resulting motion:

1. The system stayed stationary (1$^{st}$ Law), with no movement of its centre of mass;
2. The fluid shifted (its centre of mass) left because I pushed it (2$^{nd}$ Law); and
3. I, Isaac, moved right because the fluid pushed me back (reaction; 3$^{rd}$ Law)."

So far, this is all fairly straightforward, first-year physics for a multi-component system with no external forces.

**2 Fickian transport**

The second case is designed in contrast to the first, to illustrate transport with no fluid motion. Isaac is again present but now two gases, nitrous oxide ($N_2O$) and carbon dioxide ($CO_2$) compose the fluid. Each gas type occupies identical volume and are at the same temperature and pressure; hence, there are equal moles of each gas.

[Figure]

Now, if we ask Isaac about the motion of the system and its elements, he might say something quite different, along the following lines:

"Except for the mixing of $N_2O$ and $CO_2$ (about which ask Adolf Fick) this was rather boring and trivial:

1.  The system remained stationary (1$^{st}$ Law);
2.  *Since the two gases have equal molecular mass (44 g mol$^{-1}$)*, the fluid's centre of mass did not change and it experienced no force (2$^{nd}$ Law); and
3.  There was no action, and so no reaction, and I, Isaac, did not move (3$^{rd}$ Law)."

This is again straightforward, but the italics give an indication of where we are heading, into perhaps unfamiliar territory.

**3 Discerning types of transport**

The third case is designed in demonstrate the difference between kinematic and inertial points of view regarding Fick's law. It is kinematically identical to the second, but now with two gases of very different mass, hydrogen ($H_2$, 2 g mol$^{-1}$) and $Xe$ (131.3 g mol$^{-1}$).

[Figure]

Now, if we ask Isaac about the motion of the system and its elements, he might say something like the following:

"This is very similar to the first case above involving $Xe$ and the vacuum. Since 98% of the fluid mass is $Xe$, the fluid's centre of mass is initially on the right, very near to where it was at the start of Case 1, and therefore the situation is much the same:

1.  The system remained stationary (1$^{st}$ Law);
2.  The fluid shifted (its centre of mass) left because I pushed it (2$^{nd}$ Law); and
3.  I, Isaac, moved right because the fluid pushed me back (3$^{rd}$ Law)."

This no longer straightforward, regarding transport of the two gas types. The $Xe$ moved left mostly by Newtonian transport, helped a little bit by Fickian transport. The $H_2$ had to diffuse upstream in order to achieve the same overall displacement magnitude as the $Xe$, requiring large Fickian transport to the right to overcome the Newtonian transport to the left.

If the Navier-Stokes equation fails to describe the motion of the fluid due to the lack of a pressure gradient force, then we should interpret this as a shortcoming of the Navier-Stokes equation, and not of the laws that it attempts (and fails, in this case) to represent.

The key point to recognise here is that, although Case 3 is identical to Case 2 kinematically, in *inertial* terms it more closely resembles Case 1, and therefore inertia cannot be neglected when describing diffusion. Respecting Newton's laws, the determinant of diffusion is the gradient in the mass fraction, and not the molar fraction.

**4 Newtonian transport that may seem counter-intuitive**

The fourth case adds trace amounts of carbon dioxide ($CO_2$; 44 g mol$^{-1}$) to Case 3 above, specified so as to demonstrate the error of specifying Fick's law using molar fraction gradients. Again, we have the lighter gas on the left (2 g mol$^{-1}$) and the heavier gas on the right (131.3 g mol$^{-1}$), but now each side is "doped" with a tiny mass fraction (1 mg kg$^{-1}$) of $CO_2$ that negligibly influences the effective molecular mass of the mixtures. In terms of mass, we can still treat the lighter gas as hydrogen ($H_2$, 2 g mol$^{-1}$) and the heavier gas as Xe (131.3 g mol$^{-1}$), each negligibly contaminated with $CO_2$.

[Figure]

In inertial terms, this is >99.99% the same as Case 3, and would negligibly change Isaac's description of the situation. If asked about $CO_2$ transport, Isaac would likely respond that, since he pushed the fluid to the left, and the fluid shifted left, transport of $CO_2$ is explained by fluid motion: the $CO_2$ simply went with the flow, with no need to invoke diffusion. Indeed, using a mass-fraction gradient to specify Fick's law, we find that there is no diffusion in this case. However, if we convert the mass fractions to molar fractions (using molecular masses), we find that there is 0.05 ppm $CO_2$ on the left versus 2.98 ppm $CO_2$ on the right, suggesting erroneously from Roderick and Shakespeare's version of Fick's law that diffusion is responsible for $CO_2$ transport.

**Conclusion**

I believe that these cases demonstrate that the authors have used an incorrect version of Fick's law. And since their goal is to describe water vapour diffusion – from lighter, moister air towards heavier, drier air – they need to specify this correctly before addressing the complicated issue of thermodiffusion.

---

## Referee Comment (RC2)

**Roderick + Shakespeare**

This is an unusual and nicely scholarly piece of work which should be published, more or less as is.

It is interesting to discover that the Soret effect has received so little experimental investigation.  In that regard, I note that philosopher Nancy Cartwright in her book How the Laws of Physics Lie uses the Soret effect as an established example of a coupled-flux process in discussing causal inference. Perhaps less established than she thought.

I have really only two comments.

The first is to wonder if the authors might spend a little more time in discussing the boundary layer structure in evaporation:  how does the temperature vary across it?  … Can we assume that there is local kinetic (thermal) equilibrium within the boundary layer?  What are reasonable boundary layer thicknesses and temperature gradients?  There is brief mention only in lines 170-173.

The second is to ask what is the connection between the the framework/ analysis set out in this paper and the description of thermal diffusion in porous media (water, liquid and vapour) originally set out by Philip and de Vries (1957) and later papers (perhaps Luikov too around the same time).  Have I missed something here or should these analyses all be consistent?

---

## Referee Comment (RC5)

**Review of egusphere-2024-203 by Roderick and Shakespeare**

This short technical note deals with quantifying the relative contribution of the Soret effect (i.e. thermo-diffusion) compared to regular "Fickian" diffusion in the evaporation of open water bodies. The motivation behind the paper is to justify the standard practice of neglecting the Sorret effect for evaporation. For this purpose, it is based on the gas kinetic theory of Chapman and his colleagues (completed with experimental determination of the Sorret effect in binary mixtures) and on a recent dataset of evaporation rates under controlled conditions.

I think the paper is of general interest for people working of water vapor transport (even beyond the sole problem of open water evaporation) and is well suited for HESS. There is however one main point of concern (General Comment 1) that I would like the authors to clarify.

**General Comments**

1 – It is stated at the very start of the paper that the evaporation rate of an open water body is controlled by the ability of water vapor to diffuse in the air. From what I understand this is clearly the case when the air above the water is still (in this case, diffusion in the limiting factor, effectively setting its rate for the evaporation). However, as soon as wind and turbulence is involved, I have issues understanding to what extent the evaporation rate remains controlled by the molecular diffusion in the air.
I'm no boundary-layer scientist, but from what I understand, the diffusion flux of Eq. 4 applies within the micro layer (following the wording of Roland B. Stull's "An introduction to Boundary Layer Meteorology"), i.e. the zone just above the surface where molecular diffusion dominates. Thus, if it is to be translated into Eq. 5 I would say that the gradients of concentration and temperature are to be taken across this micro-layer. And I'm not sure that the concentration and temperature at the top of this micro-layer can be taken as $x_a$ and $T_a$ (as they are influenced by the surface).
Otherwise, if $x_a$ and $T_a$ are taken to define the gradients, I think the diffusivity should rather be some "effective" diffusivity (including turbulent effects) and thus does not match the molecular diffusivity anymore. And in this case, it is not clear to me that one can upscale the Soret effect to the whole boundary layer in a similar fashion (i.e. that there is an effective Sorret flux, including turbulence, that has the same form and the same thermal diffusivity ratio as in the purely molecular case).
In other words, I think the problem boils down to the difficulty of reconstructing the surface concentration and temperature gradients based on the "air" values, which I assume can be significantly different from what happen in the micro-layer.

I would thus like the authors to clarify this point. Especially, references to pre-existing literature treating this problem and relating molecular diffusion in the micro-layer to the air temperature/concentration would be appreciated.

2 – From what I understand the motivation behind this close look at the Soret effect stems from the recent study of Griffani et al. (2024), that states that thermo-diffusion can be an effective mechanism of water vapor transport and should not always be neglected. However, this motivation only appears in the discussion. I think it could be quite beneficial to include this in the introduction, as it relates to the state-of-the-art on the subject.

**Specific and technical comments**

Abstract – I would systematically say "evaporation from open water" rather than simply "evaporation".

L15 – As mentioned in the General Comment 1, references to pre-existing literature would be beneficial here.

L21 – The mention of the Duffour effect is a bit off to me, especially as it is no longer mentioned in the text. It could potentially be discussed a bit more in the Discussion and Conclusions section, notably mentioning that the Onsager reciprocal relations allows one to estimate the Duffour effect from Sorret.

L76 and L84 -  I would say "limiting conditions" rather than "boundary conditions".

L82 – If I'm not wrong the quadratic form was proposed in the work of Chapman (and certainly others). You could refer to them to justify this specific functional form.

Eq 5 - It might be just me, but I'm not fond of mixing alphabetical and digits in Equations, as it obscure physical variables from actual math constant (and I find it harder to read and interpret). I would keep alpha_T rather than 0.05.

L136 – It relates to General comment 1. Could you elaborate on the physical significance of Delta z? Is it the thickness of the micro-layer (where diffusion dominates)?

Eqs 6a and 6b – Same as Eq. 5. I would go straight to the point and say that Eq 5 using some standards values yields a 99.6%/0.4% partition for the Fickian and Sorret fluxes.

L149 – For me, "vanishingly small" implies that the Sorret contribution strictly goes to zero when xs equals xa. However, I do not think it's the case as small Sorret contribution remains non zero (if xs is different from 0 or 1, and Ts different from Ta).

L155 – If it's the air temperature, please use Ta rather than T.

Figure 2 – I find it hard to determine the relative contribution of the Sorret flux in panel c near the origin of the graph. Perhaps add a second y-scale the relative contribution of the Sorret flux as a scatter of the total evaporation flux.

L194 – I think the wording could be improved. From what I understand, the issue is that Griffani et al.'s work is based on Landau's derivation which assumes that (ii) the water vapor molecule are much lighter than the dry air and (ii) the collisions are elastic; both assumption not applying to the actual mixture of water and dry air.
The current wording rather suggests that the issues are (i) Griffani et al. is based on Landau (M=1 and elastic collisions) and (ii) the collisions are assumed elastic; in which the same issue actual appears twice.

L209 – Taking alpha_t as 0.05 rather than 0.5 effectively reduces the Sorret flux by an order of magnitude, and the 30% contribution of Sorret in Griffani et al. now becomes a 3% contribution, which is deemed negligible. That being said, I would still leave the door open that in almost all natural conditions, the Sorret effect appears negligible, but that pathological cases (i.e. very high temperature gradients with little concentration gradients) can still exist (and it links with the paragraph below).

---

## Referee Comment (RC6)

**Review report on "Technical note: An assessment of the relative contribution of the Soret effect to open water evaporation"**

by Demetris Koutsoyiannis

| | |
|---|---|
| Author(s): | Michael L. Roderick and Callum J. Shakespeare (Australia) |
| Journal: | *HESS* |
| Journal's Ref.: | egusphere-2024-2023 (HESS) |
| Reviewer's Ref.: | DK-JR-387 |
| Date: | 2024-09-26 |
| Recommendation: | Accept |

**Reviewer's assertion**: It is my opinion that a shift from anonymous to eponymous (signed) reviewing would help the scientific community to be more cooperative, democratic, equitable, ethical, productive and responsible. Therefore, it is my choice, consistent with my aesthetic attitude, to sign my reviews. Furthermore, I believe that the current trend in the review system to seek credit for anonymous transactions (by asking recognition for anonymous reviews through Web of Science, a practice also encouraged by journals) is problematic on ethical and aesthetic grounds. Only eponymous transactions can deserve recognition.

After the introduction of chatbots, which can produce automatic reviews superior that the typical average review, I believe that the peer-review system needs a major overhaul on the basis:

**TEAR**: Transparency, Eponymity, Accountability, Responsibility.

I am not an expert on thermodiffusion and the Soret effect. Rather, I accepted the review invitation to learn, as a student, about an issue that I did not know before. I am very satisfied as the authors, as well as the other reviewers, are indeed so very knowledgeable that I did learn a lot.

I see that the other reviewers have made several constructive suggestions, which the authors responded to, and I feel there is no need for me to make additional comments. One unaddressed issue is Kowalski's point that the authors' "specification of Fick's 1st Law is incorrect" and that "Eq. (1) is dimensionally inhomogeneous unless the diffusive flux density ($J$) is specified in molar terms". By the way, I enjoyed Kowalski's examples, yet I do not see his contradiction of "Newtonian transport" vs "Fickian transport". I believe the former needs to be complemented by the latter (which reflects the principle of maximum entropy) to explain the phenomenon fully. Otherwise, the motion of the molecules from one half of the container to the other half does not make sense.

Feeling like a student, I did my homework on this issue, which I include as an Appendix to my review. The results of my homework show that the mass description is precisely equivalent to the molar description and that the only change needed to the paper is to correct the phrase (above Eq. (1)) "diffusive flux $J$ (kg m$^{-2}$ s$^{-1}$)" to "diffusive flux

$J$ (mol m$^{-2}$ s$^{-1}$). Otherwise Eq. (1) would indeed be dimensionally inhomogeneous. I hope the authors and discussers find my homework correct.

**Appendix**

Fick's law (in mass units, as I used to teach it, in one dimensional form for simplicity) is:

$$J = -D \frac{\mathrm{d}\rho_v}{\mathrm{d}z}$$

with

$J$ [kg m$^{-2}$s$^{-1}$]: the water vapour flux,
$D$ [m$^2$s$^{-1}$]: the diffusion coefficient,
$\rho_v$ [kg m$^{-3}$]: the density of water vapour, and
$z$ [m] : the vertical coordinate.

Now, the densities of water vapour and of the mixture are, respectively:

$$\rho_v = \frac{M_v}{V}, \qquad \rho = \frac{M_{\mathrm{TOT}}}{V}$$

where $M_v$ [kg] and $M_{\mathrm{TOT}}$ [kg] are the masses of the water vapour and the mixture at a specified volume $V$ [m$^3$]. From these we get

$$\rho_v = \frac{M_v}{M_{\mathrm{TOT}}}\rho = \frac{m_v n_v}{m_{\mathrm{TOT}} n_{\mathrm{TOT}}}\rho = \frac{m_v}{m_{\mathrm{TOT}}}x\rho$$

where $m_v$ [kg mol$^{-1}$] and $m_{\mathrm{TOT}}$ [kg mol$^{-1}$] are the respective molar masses, $n_v$ [mol] and $n_{\mathrm{TOT}}$ [mol] the respective number of moles, and $x := n_v/n_{\mathrm{TOT}}$ the mole fraction of water vapour in the mixture.

Hence

$$J = -\rho \frac{m_v}{m_{\mathrm{TOT}}} D \frac{\mathrm{d}x}{\mathrm{d}z}$$

Now, if we define the molar density of the mixture

$$\rho_{\mathrm{mol}} = \frac{\rho}{m_{\mathrm{TOT}}}, \qquad \rho_{\mathrm{mol}} : [\mathrm{mol\ m}^{-3}]$$

and the molar water vapour flux

$$J_{\mathrm{mol}} = \frac{J}{m_v}, \qquad J_{\mathrm{mol}} : \left[ \frac{\mathrm{kg\ m}^{-2}\mathrm{s}^{-1}}{\mathrm{kg\ mol}^{-1}} = \mathrm{mol\ m}^{-2}\mathrm{s}^{-1} \right]$$

we get

$$J_{\mathrm{mol}} = -\rho_{\mathrm{mol}} D \frac{\mathrm{d}x}{\mathrm{d}z}$$

which is the first term in Equation (1) in Roderick and Shakespeare.

---

## Referee Comment (RC7)

**Reply to Demetris Koutsoyiannis**

I agree with the assertion regarding the benefits of signing reviewer comments, and applaud the attitude of getting involved with the outlook of a student. I want to think that I use the same outlook, although I am aware that my writing can sometimes be taken as overly authoritative and even offensive to some readers. I assure that it is not my intent to offend, but rather to learn the truth, and to defend the truth when I believe that I have found it.

With that in mind, I cannot agree with Professor Koutsoyiannis in his assertion that "the mass description is precisely equivalent to the molar description", because I believe his derivation contains two incorrect assumptions. If I follow correctly the Appendix, just prior to the word "Hence", Professor Koutsoyiannis's version of Fick's law could be written, via substitution, as:

$$J = -D \frac{d}{dz}\left[\frac{m_v}{m_{TOT}} x\rho\right]$$

So that simplification to the version just following the word "Hence" requires two invalid assumptions:

1. that $m_{TOT}$ is constant (and so can be extracted from the derivative), but molar mass varies with humidity and so $\frac{d}{dz}[m_{TOT}] \neq 0$; and
2. that $\rho$ is constant (and so can be extracted from the derivative), but $\rho$ varies according to various factors*, principally the temperature ($T$), and so $\frac{d\rho}{dz} \neq 0$. Given that the issue at hand is the role of $T$ gradients in modifying Fickian diffusion, or Soret effect, such an assumption is particularly inappropriate.

It is my contention, based on applying Newton's laws to the mixing *and inertia* of fluids with different molar masses, that the correct version of Fick's law must be

$$J = -\rho D \frac{df_v}{dz}$$

where $f_v$ is the mass fraction, also known as the specific humidity ($q$).

*The other factors influencing gradients in $\rho$ are the pressure (via the ideal gas law) and the humidity.

---

## Author Comment (AC2)

**HESS manuscript:** https://doi.org/10.5194/egusphere-2024-2023

**Title: Technical note: An assessment of the relative contribution of the Soret effect to open water evaporation**

**Authors: Roderick ML & Shakespeare CJ**

**Response to RC3 (Anon. Reviewer 3, 18 Sep 2024)**

*Review comments in italics.*

**Author Response in bold.**

1. *This manuscript gives a detailed discussion on the relative contributions of Fickian's diffusion and Soret effect on open water evaporation, and justifies the popular practice of estimating open water evaporation through the water vapor concentration gradients. The authors prove that the Soret effect is two orders of magnitude smaller than that by concentration-dependent diffusion. I consider such kind of work is just quite rare because of limited laboratory experiments and it is valuable for us to understand the processes behind it.*

**We thank the reviewer for the comments.**

2. *In figure1, the thermal diffusion factor is 0.05 for N2-N2O and N2-CO2, but 0.33 for H2-CO2. It seems that even though the thermal diffusion factor of 0.33 is used for Soret effect estimation, its contribution is still much smaller compared to the concentration gradients.*

**We assume the reviewer means that even if we used the (incorrect) larger value of 0.33 (instead of 0.05) the conclusions would not be materially altered. If so, we agree.**

3. *In abstract, the authors mentioned that "under typical environmental conditions it is at least two orders of magnitude smaller than classical concentration-dependent mass ('Fickian') diffusion.". From Figure1 and Figure 2, we could find that the Soret effect can be neglected in open water estimation. I 'm just wandering under what conditions in open water evaporation estimation (fresh water and saline water) the Soret effect can not be ignored? If no, please add some examples in the discussion of the manuscript.*

Adopting a value of 0.05 for the thermal diffusion factor ($\alpha_T$) we can pose an answer to the reviewers question about when the Soret effect might be important. The obvious condition would be where the absolute magnitude of the surface to air $T$ difference was much larger. We assume the same data as in Table 3 (air at 298 K and 60% relative humidity, atmospheric pressure of 1 bar) but now we increase the surface temperature to 372 K (just under the boiling point of liquid water). At face value the relative magnitude of the Soret effect would be larger but in fact it is smaller than our original calculation (now 0.2 % of the total flux instead of 0.4%, based on Eqn 6a). The reason is that the surface is assumed saturated as appropriate for open water. Hence when the surface-air $T$ difference is larger so is the gradient in mole fraction. The only way we could envisage a significant Soret effect is for the open water to be replaced by a highly concentrated brine-type solution where the saturated vapour pressure at the surface is close to zero and does not increase (much) with change in $T$. Again we could manufacture such a scenario but the absolute magnitude of the evaporative flux would be small. We can mention this important point in a future revision of the manuscript.

4.  *Further, the figure quality is not good, please make figures with good quality.*

The manuscript was prepared using 'cut and paste' to insert the figures from a PDF into a Word document and we may have inadvertently used a low resolution during that process. However, for the final printed version the figures will be high quality PDFs. We will also increase the font size on the axis labels as per comments by another review (Kowalski).

5.  *Generally, I consider the manuscript is a good materials that can help us to understand clear the water-atmosphere interaction processes.*

We thank the reviewer for the comments.

---

## Author Comment (AC3)

**HESS manuscript:** https://doi.org/10.5194/egusphere-2024-2023

**Title: Technical note: An assessment of the relative contribution of the Soret effect to open water evaporation**

**Authors: Roderick ML & Shakespeare CJ**

**Response to RC4 (Anon. Reviewer 4, 23 Sep 2024)**

*Review comments in italics.*

**Author Response in bold.**

1. *This study quantifies the magnitude of the Soret effect on open water evaporation and demonstrate that it is typically two orders of magnitude smaller than the mass diffusion component (Fickian diffusion). This finding justifies the common practice of ignoring the Soret effect when describing evaporation in hydrological sciences.*

   *I believe this is an important study that should be accepted after minor corrections and clarifications. The manuscript is well-written and exhibits excellent readability. However, it may benefit from clarifications regarding the following points.*

   **We thank the reviewer for the thoughtful comments.**

*Comments:*

2. *C1: Multiple times throughout the manuscript, the sentences give an impression that evaporation is entirely a Fick's diffusion process (Line 26,212). However, estimation of evaporation also requires an explicit consideration of an energy term. Over open-water surfaces the gradient in the water-vapor is further strongly controlled by changes in temperature and incoming energy as reflected in the classical equilibrium energy partitioning approach (Slatyer and McIlroy, 1961).*

   **The energy balance approach (e.g. Slatyer & McIllroy 1961) is an alternative to the mass transfer approach (e.g. Fick's law) and this has been widely used in hydrology and agriculture (also see Yang & Roderick 2019, QJRMS, 145, 1118-1129). However, it has been held for more than 200 years (since Dalton's 1802 paper) that evaporation can be specified solely using a mass transfer approach.**

   **Energetic constraints are needed in addition to (but do not replace) the mass transfer formulation for evaporation if one seeks to model the evolution of the system over time (e.g., in a land surface, atmosphere or ocean model). Here, however, we have direct experimental control of all key variables (air temperature/humidity and windspeed) and measure evaporation and surface temperature directly which avoids the need for such modelling.**

3. *C2: In Line 123, it may be useful to provide a sensitivity estimate of α_T with respect to temperature using the equation from Youssef et al. (1965). This would demonstrate that variations in α_T with temperature are not substantial enough to cause significant changes in the Soret effect.*

**We have done that on line 122 where we note that $\alpha_T = 0.05$ (at $T = 328$ K, Fig. 1b) would become $\alpha_T = 0.048$ at $T = 300$ K using the Youssef et al (1965, their Eqn 7) results.**

4. *C3: In Line 135, Check the equation. Should x_a be written as a dependence on T_a as well. (x_s(T_s) + x_a(T_a))/2.*

**The mole fraction in the air is specified directly by measurement (i.e., $x_a$) while that for the surface ($x_s$) is calculated at the surface temperature $T_s$ which assumes that $x_s$ is a direct function of $T_s$ and this direct dependence is denoted using $x_s(T_s)$.**

5. *C4: Line 147: The authors quantified the relative contribution of the Soret effect for standard conditions with data described in Table 3. Later they talk about describing the condition where mole fraction gradient would become zero and Soret effect would then be 100% of the total flux. They mention that this leads to total flux being vanishingly small "as described below (line 147)". However, the results for this condition are not described unless they are referring to the next section of the manuscript.*

**We did not understand the point. The sentence starting on L147 reads; reads "However, that total flux would be vanishingly small as we show below."; so we are referring directly to the following results.**

6. *C5: Line 171: It may be helpful to add a brief discussion for why the boundary layer thickness declines with wind-speed for a wider audience.*

**That has been fully described in the cited reference (Lim et al 2012) and in our opinion is beyond the scope of this work.**

7. *C6: One key difference between Griffani et al. (2024) and this study is the magnitude of the thermal diffusion factor, which is one order of magnitude higher in the former. While the authors provide a thoughtful justification for their use of 0.05 for the magnitude of diffusion coefficient, there is no experimental data for H2O-dry air mixtures. It would be important to validate these estimates with new experiments; perhaps the authors could include this as an outlook for future research.*

**Good point. It is truly astonishing that we could not locate a single experiment involving water vapor despite a search lasting a few weeks across the libraries of the world. We can add that point in the discussion as suggested.**

---

## Author Comment (AC4)

**HESS manuscript:** https://doi.org/10.5194/egusphere-2024-2023

**Title:** **Technical note: An assessment of the relative contribution of the Soret effect to open water evaporation**

**Authors:** **Roderick ML & Shakespeare CJ**

**Response to RC5 (Anon. Reviewer 5, 23 Sep 2024)**

*Review comments in italics.*

**Author Response in bold.**

1. *This short technical note deals with quantifying the relative contribution of the Soret effect (i.e. thermodiffusion) compared to regular "Fickian" diffusion in the evaporation of open water bodies. The motivation behind the paper is to justify the standard practice of neglecting the Sorret effect for evaporation. For this purpose, it is based on the gas kinetic theory of Chapman and his colleagues (completed with experimental determination of the Sorret effect in binary mixtures) and on a recent dataset of evaporation rates under controlled conditions.*

   *I think the paper is of general interest for people working of water vapor transport (even beyond the sole problem of open water evaporation) and is well suited for HESS. There is however one main point of concern (General Comment 1) that I would like the authors to clarify.*

**We thank the reviewer for the comments.**

2. *General Comments 1 – It is stated at the very start of the paper that the evaporation rate of an open water body is controlled by the ability of water vapor to diffuse in the air. From what I understand this is clearly the case when the air above the water is still (in this case, diffusion in the limiting factor, effectively setting its rate for the evaporation). However, as soon as wind and turbulence is involved, I have issues understanding to what extent the evaporation rate remains controlled by the molecular diffusion in the air. I'm no boundary-layer scientist, but from what I understand, the diffusion flux of Eq. 4 applies within the micro layer (following the wording of Roland B. Stull's "An introduction to Boundary Layer Meteorology"), i.e. the zone just above the surface where molecular diffusion dominates. Thus, if it is to be translated into Eq. 5 I would say that the gradients of concentration and temperature are to be taken across this micro-layer. And I'm not sure that the concentration and temperature at the top of this micro-layer can be taken as xa and Ta (as they are influenced by the surface). Otherwise, if xa and Ta are taken to define the gradients, I think the diffusivity should rather be some "effective" diffusivity (including turbulent effects) and thus does not match the molecular diffusivity anymore. And in this case, it is not clear to me that one can upscale the Sorret effect to*

*the whole boundary layer in a similar fashion (i.e. that there is an effective Sorret flux, including turbulence, that has the same form and the same thermal diffusivity ratio as in the purely molecular case). In other words, I think the problem boils down to the difficulty of reconstructing the surface concentration and temperature gradients based on the "air" values, which I assume can be significantly different from what happen in the micro-layer. I would thus like the authors to clarify this point. Especially, references to pre-existing literature treating this problem and relating molecular diffusion in the micro-layer to the air temperature/concentration would be appreciated.*

**We thank the reviewer for this comment. The reviewer is correct in that we are referring to the molecular boundary layer – what they term the "micro-layer". Experiments (see Doe, 1967, Measurement of a mass transfer boundary layer. Nature 216, 1101–1103, doi:10.1038/2161101a0) show that that vapour concentration across this layer changes from the surface value (saturated, $x_s$) to the free stream value ($x_a$) as has been assumed in our formulation. Indeed, our formulation follows the standard description of the so-called threshold model that has been in use in hydrology, agriculture and climate for the last century. The details are fully described by Fig. 1 in the cited reference (Lim et al, 2012) which is reproduced below:**

W.H. Lim et al. / Agricultural and Fc

**Fig. 1.** A schematic diagram of the "threshold model" adopted here for the variation in vapour pressure ($e$) with height ($z$) above the evaporating surface. After Leighly (1937) and Machin (1964, 1970).

**Fig. R1 Reproduced from Lim et al (2012)**

**The approach is to replace the actual profile (full line in Fig. R1) with an assumed "threshold-type" model (dotted lines in Fig. R1).**

3. *2 – From what I understand the motivation behind this close look at the Sorret effect stems from the recent study of Griffani et al. (2024), that states that thermo-diffusion can*

*be an effective mechanism of water vapor transport and should not always be neglected. However, this motivation only appears in the discussion. I think it could be quite beneficial to include this in the introduction, as it relates to the state-of-the-art on the subject.*

**We accept the point and we can add a reference to Griffani et al as well as the necessary text in the introduction.**

*Specific and technical comments*

4. *Abstract – I would systematically say "evaporation from open water" rather than simply "evaporation".*

**On L113 we can modify the text from "… by assuming evaporation follows …." To read "… by assuming evaporation from open water follows …."**.

5. *L15 – As mentioned in the General Comment 1, references to pre-existing literature would be beneficial here.*

**We can add a reference to the classical Monteith and Unsworth 2008 textbook, e.g., …. gradient (Fick's Law) (Monteith and Unsworth, 2008) as requested.**

6. *L21 – The mention of the Duffour effect is a bit off to me, especially as it is no longer mentioned in the text. It could potentially be discussed a bit more in the Discussion and Conclusions section, notably mentioning that the Onsager reciprocal relations allows one to estimate the Duffour effect from Sorret.*

**We agree that the Dufour effect is left hanging on it's own. We just wanted to make the point that the Onsager-based coupling leads to other effects. We would prefer to leave the text as it is there to highlight that additional flux which we do not study.**

7. *L76 and L84 - I would say "limiting conditions" rather than "boundary conditions".*

**Ok, we can do that.**

8. *L82 – If I'm not wrong the quadratic form was proposed in the work of Chapman (and certainly others). You could refer to them to justify this specific functional form*

**Yes, Chapman did propose that. We can add the reference as requested.**

9. *Eq 5 - It might be just me, but I'm not fond of mixing alphabetical and digits in Equations, as it obscure physical variables from actual math constant (and I find it harder to read and interpret). I would keep alpha_T rather than 0.05.*

**We accept the point that this is a matter of "taste". We also prefer using symbols but in this case we have supplied the actual numeric value that is assumed unchanging (see L131).**

*10. L136 – It relates to General comment 1. Could you elaborate on the physical significance of Delta z? Is it the thickness of the micro-layer (where diffusion dominates)?*

**Yes, it is the thickness you mentioned. We can modify the text leading to Eqn 5 (on existing lines 131-132) by stating that we follow the classical threshold model where the temperature and water vapour follow a linear profile from the (saturated) surface to the free stream value in the air over the distance $\Delta z$. (Also see response to point 2 above.)**

*11. Eqs 6a and 6b – Same as Eq. 5. I would go straight to the point and say that Eq 5 using some standards values yields a 99.6%/0.4% partition for the Fickian and Sorret fluxes.*

**Again a matter of taste. We think that having the numbers written out makes it very clear that the Soret effect will be small (by at least two orders) regardless of the boundary layer model chosen.**

*12. L149 – For me, "vanishingly small" implies that the Sorret contribution strictly goes to zero when xs equals xa. However, I do not think it's the case as small Sorret contribution remains non zero (if xs is different from 0 or 1, and Ts different from Ta).*

**Good point. We can modify the text accordingly.**

*13. L155 – If it's the air temperature, please use Ta rather than T.*

**Good point. It is the air temperature and we can modify the text to read; "… air T range from 15 ….".**

*14. Figure 2 – I find it hard to determine the relative contribution of the Sorret flux in panel c near the origin of the graph. Perhaps add a second y-scale the relative contribution of the Sorret flux as a scatter of the total evaporation flux.*

**We agree that the Soret flux (denoted $E_T$) is hard to precisely determine near the origin but the main point here is not the exact numerical value, but rather that it is very small.**

*15. L194 – I think the wording could be improved. From what I understand, the issue is that Griffani et al.'s work is based on Landau's derivation which assumes that (ii) the water vapor molecule are much lighter than the dry air and (ii) the collisions are elastic; both assumption not applying to the actual mixture of water and dry air. The current wording rather suggests that the issues are (i) Griffani et al. is based on Landau (M=1 and elastic*

*collisions) and (ii) the collisions are assumed elastic; in which the same issue actual appears twice.*

**We can try and improve the wording as suggested. We note that the interpretation of the reviewer is correct.**

16. *L209 – Taking alpha_t as 0.05 rather than 0.5 effectively reduces the Sorret flux by an order of magnitude, and the 30% contribution of Sorret in Griffani et al. now becomes a 3% contribution, which is deemed negligible. That being said, I would still leave the door open that in almost all natural conditions, the Sorret effect appears negligible, but that pathological cases (i.e. very high temperature gradients with little concentration gradients) can still exist (and it links with the paragraph below).*

**Reviewer 3 raised a similar sentiment (see point 3 in the response to Reviewer 3). We repeat that response here:**

**Adopting a value of 0.05 for the thermal diffusion factor ($\alpha_T$) we can pose an answer to the reviewers question about when the Soret effect might be important. The obvious condition would be where the absolute magnitude of the surface to air *T* difference was much larger. We assume the same data as in Table 3 (air at 298 K and 60% relative humidity, atmospheric pressure of 1 bar) but now we increase the surface temperature to 372 K (just under the boiling point of liquid water). At face value the relative magnitude of the Soret effect would be larger but in fact it is smaller than our original calculation (now 0.2 % of the total flux instead of 0.4%, based on Eqn 6a). The reason is that the surface is assumed saturated as appropriate for open water. Hence when the surface-air *T* difference is larger so is the gradient in mole fraction. The only way we could envisage a significant Soret effect is for the open water to be replaced by a highly concentrated brine-type solution where the saturated vapour pressure at the surface is close to zero and does not increase (much) with change in *T*. Again we could manufacture such a scenario but the absolute magnitude of the evaporative flux would be small. We can mention this important point in a future revision of the manuscript.**

---

## Author Comment (AC5)

HESS manuscript: https://doi.org/10.5194/egusphere-2024-2023

**Title:** Technical note: An assessment of the relative contribution of the Soret effect to open water evaporation

**Authors:** Roderick ML & Shakespeare CJ

**Combined Response to:**

      **RC6 (Demetris Koutsoyiannis, 26 Sep 2024)**

      **RC7 (Andrew Kowalski, 27 Sep 2024)**

      **RC8 (Demetris Koutsoyiannis, 27 Sep 2024)**

**Author Response in bold.**

**The review (RC6) did not raise any issues (other than the units typo also identified by Dr Kowalski). We thank Dr Koutsoyiannis for his careful analysis of the mass vs molar based expressions for diffusion (RC6).**

**In a further response, Dr Kowalski (RC7) has argued that the mass vs molar-based derivation by Dr Koutsoyiannis (RC6) made an error and that the density cannot be taken inside the derivative which has subsequently been noted (RC8).**

**In summary, if the density were to be placed outside the derivative then one can still convert between mass and molar based expressions.**

**To pursue this topic further we consulted a standard engineering reference on the topic, i.e., the CRC Mechanical Engineering Handbook (Kreith et al 1999) and the relevant "snapshot" from that text is shown below (Fig. R1). This handbook firstly asserted that Fick's Law can be specified on either a mass or molar basis and that both expressions are equivalent. Second, the text agreed with the point of view of Dr Kowalski that the density must be outside the integral (as we had done in our submitted manuscript).**

**We also read widely on the topic and agree with Dr Kowalski that diffusion has been described in many ways over the years. In practice the expression using density inside the derivative (as used in RC7) is commonly used in liquids and solids without much error. However, in a gas the kinetic theory predicts that the density should be outside the derivative.**

**We thank the reviewers (Kowalski, Koutsoyiannis) for forcing us to look more into this issue than we had done previously.**

**Mechanisms of Diffusion**

**Ordinary Diffusion**

Fick's law of ordinary diffusion is a linear relation between the rate of diffusion of a chemical species and the local concentration gradient of that species. It is exact for a binary gas mixture, for which the kinetic theory of gases gives

$$j_1 = -\rho \mathcal{D}_{12} \nabla m_1 \ \text{kg/m}^2 \ \text{sec} \tag{4.7.24a}$$

© 1999 by CRC Press LLC

on a mass basis, and

*Typo.*

*Mol* $m^{-2} s^{-1}$

$$J_1^* = -c\mathcal{D}_{12} \nabla x_1 \ \text{kg/m}^2\text{sec} \tag{4.7.24b}$$

on a molar basis; $\mathcal{D}_{12}$ (m²/sec) is the binary diffusion coefficient (or mass diffusivity), and $\mathcal{D}_{21} = \mathcal{D}_{21}$. Equations (4.7.24a) and (4.7.24b) are mathematically equivalent; however, notice that it is incorrect to write

$$j_i = -\mathcal{D}_{12} \nabla \rho_1 \tag{4.7.25}$$

since $\nabla \rho_1 \neq \rho \nabla m_1$ in general. Fick's law in the form of Equations (4.7.24a) and (4.7.24b) is also valid for dilute liquid and solid solutions, for which it is often possible to assume $\rho$ (or $c$) constant, and then Equation (4.7.25) or its molar equivalent are good approximations.

**Fig. R1** "Snapshot" of part of pages 4-211 and 4-212 from Kreith et al (1999).

**References**

Kreith, F., Boehm, R., Raithby, G., Hollands, K., Suryanarayana, N., Modest, M., VP, V., Chen, J., Lior, N., Shah, R., Bell, K., Moffat, R., Mills, A., Bergles, A., Swanson, L., Antonetti, V., Irvine Jr, T., and Capobianchi, M.: Heat and Mass Transfer, in: Mechanical Engineering Handbook, edited by: Kreith, F., CRC Press LLC, Boca Raton, 1999.

---

## Author Comment (AC6)

**HESS manuscript:** https://doi.org/10.5194/egusphere-2024-2023

**Title:** Technical note: An assessment of the relative contribution of the Soret effect to open water evaporation

**Authors:** Roderick ML & Shakespeare CJ

**Response to RC1 (Dr Andrew Kowalski, 3 Sep 2024)**

*Review comments in italics.*

**Author Response in bold.**

**Given the nature of the review we have chosen to reverse the order of the questions to be addressed.**

*2. Independent of this, I point out that the authors' Eq. (1) is dimensionally inhomogeneous unless the diffusive flux density (J) is specified in molar terms, with units as in Table 1 rather than the mass-based units that they indicate at line 62. Also, the axis labels should be larger in order to be legible, particularly for Figure 2.*

**Yes, there was a 'typo' on line 62 where the units should have been molar. Thank you.**

**We can make the text larger on the axes of both Figures 1 and 2 as suggested in a revised submission.**

*1. The manuscript by Roderick and Shakespeare purports to characterise the influence of the Soret effect, whereby temperature gradients influence mass diffusion, versus the classical concentration-dependent mass ('Fickian') diffusion. But in order to do this requires first correctly characterising Fickian diffusion, and this I believe the authors have not yet done. In brief, the authors have specified Fick's law based on gradients in the molar fraction, whereas Newtonian analyses demonstrate that it must be specified in terms of the mass fraction, and the difference between the two is hardly trivial for fluids of varying molecular mass. Respectfully, I therefore believe that the manuscript should be rejected. My arguments for why their specification of Fick's 1$^{st}$ Law is incorrect are laid out in an open-access paper (Kowalski et al., 2021) that can be accessed here (https://link.springer.com/article/10.1007/s10546-021-00605-5; see sections 3.2 and 4 in particular), but are reinforced in the attached PDF file.*

**We thank the reviewer for their considered response and for the additional document (i.e., the uploaded PDF) based on "Isaac" which we enjoyed reading. We have also carefully**

read the cited 2021 Boundary Layer Meteorology article as well as the earlier 2017 article in AtmChemPhys.

We were somewhat surprised by the ferocity (i.e., recommend complete rejection) given the highly favourable comments by the other six reviewers of the manuscript. With that background we have taken the comment seriously. In fact this was the first comment posted on our manuscript (on 3 Sep 2024) and one of us (MLR) has spent nearly all of the available time since then undertaking additional reading/research to seriously address the comment. Our combined knowledge of "diffusion" has increased substantially and for that we thank the reviewer. The extra work was worth it in this case.

The underlying basis of the assertion that we MUST use a mass-based framework has two separate points. The first is simply that one has to use a mass-based framework and the second is that we have ignored a small advective flux (i.e., the so-called Stefan flow). For the latter we accept the criticism and we intend to follow the suggestion of the reviewer by modifying Eqn 5 to include the additional advective flux (accounting for the "the bulk flow") that is implicitly requested by the reviewer. This will account for roughly 2% of the total flux and we note that it is actually slightly larger than the Soret effect. We will follow the standard mechanical engineering (Kreith et al 1999) and chemical engineering (Cussler 2009) texts when implementing this "bulk flow" based effect. The net effect of this change is that the original conclusions of the manuscript will be unaltered.

The reviewer has also asserted that diffusion can only be described using a mass-based framework. In our case the molar-based framework is useful because it fits in with recent existing work that also used a molar-based framework (Griffani et al 2024). Through the additional work we have found that it has long been established that one can use either mass- or molar-based frameworks interchangeably. There is a vast scientific literature on this topic (e.g. Cullinan 1965; Brady 1975; Miller 1986) that establishes the complete equivalence of mass- and molar-based frameworks for describing diffusion. On our reading we have found the key thing is not actually the units used (i.e., mass or molar) but instead it is the definition of the reference velocity. This key point is explained in detail in the above cited references (e.g. Cullinan 1965; Brady 1975; Miller 1986). We also identified a very succinct and elegant tabular summary of this very point in the famous Cussler textbook on diffusion (Cussler 2009: Table 3.2-1) that is reproduced below in Fig. R1. Interestingly, in Hydrology (and other climate related fields) we usually specify the diffusion coefficient of water in air based on laboratory experiments. It turns out that those experimental results are actually based on a volume-based reference velocity as is most gas-based analysis.

Table 3.2-1 *Different forms of the diffusion equation*

| Choice | Total flux (diffusion + convection) | Diffusion equation | Reference velocity | Where best used |
|---|---|---|---|---|
| Mass | $n_1 = j_1^m + \rho_1 v$ | $j_1^m = \rho_1(v_1 - v)$ $= -D\rho\nabla\omega_1$ | $v = \omega_1 v_1 + \omega_2 v_2$ $\rho v = n_1 + n_2$ | Constant-density liquids; coupled mass and momentum transport |
| Molar | $n_1 = j_1^* + c_1 v^*$ | $j_1^* = c_1(v_1 - v^*)$ $= -Dc\nabla y_1$ | $v^* = y_1 v_1 + y_2 v_2$ $cv^* = n_1 + n_2$ | Ideal gases where the total molar concentration $c$ is constant |
| Volume | $n_1 = j_1 + c_1 v^0$ | $j_1 = c_1(v_1 - v^0)$ $= -D\nabla c_1$ | $v^0 = c_1\bar{V}_1 v_1 + c_2\bar{V}_2 v_2$ $= \bar{V}_1 n_1 + \bar{V}_2 n_2$ | Best overall; good for constant-density liquids and for ideal gases; may use either mass or mole concentration |
| Solvent | $n_1 = j_1^{(2)} + c_1 v_2$ | $j_1^{(2)} = c_1(v_1 - v_2)$ $= -D_1\nabla c_1$ | $v_2$ | Rare except for membranes; note that $D_1 \neq D_2 \neq D$ |
| Maxwell–Stefan | | $\nabla y_1 = \dfrac{y_1 y_2}{D'}(v_2 - v_1)$ | None | Written for ideal gases; difficult to use in practice |

**Figure R1     Reproduction of Table 3.2-1 from Cussler (2009: p. 60) showing the equivalence of different forms of the diffusion equation.**

**Finally, we refer back to the 'Issac' examples provided by the reviewer. These involve two different gases (of different molecular mass) and what happens when they mix. This problem has been explicitly dealt with by Cussler (2009: p. 58-59) and Cussler agrees with the reviewer that that centre of mass will change. But Cussler also points out that the volume (and mole) average velocity is zero and so this is the easiest diffusion equation to use and is recommended for that reason which refutes the assertion by the reviewer. We have included "snapshots" from p. 58-59 of Cussler in an appendix to this response.**

*References*

**Brady, J. B.: Reference frames and diffusion coefficients, American Journal of Science, 275, 954-983, doi.org/10.2475/ajs.275.8.954, 1975.**

**Cullinan, H. T., Jr.: Analysis of Flux Equations of Multicomponent Diffusion, Industrial & Engineering Chemistry Fundamentals, 4, 133-139, 10.1021/i160014a005, 1965.**

**Cussler, E. L.: Diffusion: Mass Transfer in Fluid Systems, 3rd ed., Cambridge University Press, Cambridge, UK2009.**

*Kowalski, A. S., Serrano-Ortiz, P., Miranda-García, G., and Fratini, G., 2021. "Disentangling turbulent gas diffusion from non-diffusive transport in the boundary layer." Boundary-Layer Meteorology, **179** (3), 347-367.*

**Kowalski, A. S.: The boundary condition for vertical velocity and its interdependence with surface gas exchange, Atmos. Chem. Phys., 17, 8177-8187, 10.5194/acp-17-8177-2017, 2017.**

**Kreith, F., Boehm, R., Raithby, G., Hollands, K., Suryanarayana, N., Modest, M., VP, V., Chen, J., Lior, N., Shah, R., Bell, K., Moffat, R., Mills, A., Bergles, A., Swanson, L., Antonetti, V., Irvine Jr, T., and Capobianchi, M.: Heat and Mass Transfer, in: Mechanical Engineering Handbook, edited by: Kreith, F., CRC Press LLC, Boca Raton, 1999.**

**Miller, D. G.: Some comments on multicomponent diffusion: negative main term diffusion coefficients, second law constraints, solvent choices, and reference frame transformations, The Journal of Physical Chemistry, 90, 1509-1519, 10.1021/j100399a010, 1986.**

**Appendix – Snapshots of pages 58 and 59 from Cussler (2009)**

In more exact terms, we define the total mass flux $\boldsymbol{n}_1$ as the mass transported per area per time relative to fixed coordinates. This flux, in turn, is used to define an average solute velocity $\boldsymbol{v}_1$:

$$\boldsymbol{n}_1 = c_1 \boldsymbol{v}_1 \tag{3.1-2}$$

where $c_1$ is the local concentration. We then divide $\boldsymbol{v}_1$ into two parts:

$$\boldsymbol{n}_1 = c_1(\boldsymbol{v}_1 - \boldsymbol{v}^a) + c_1 \boldsymbol{v}^a = \boldsymbol{j}_1^a + c_1 \boldsymbol{v}^a \tag{3.1-3}$$

where $\boldsymbol{v}^a$ is some convective "reference" velocity. The first term $\boldsymbol{j}_1^a$ on the right-hand side of this equation represents the diffusion flux, and the second term $c_1 \boldsymbol{v}^a$ describes the convection.

Interestingly, there is no clear choice for what this convective reference velocity should be. It might be the mass average velocity that is basic to the equations of motion, which in turn are a generalization of Newton's second law. It might be the velocity of the solvent, because that species is usually present in excess. We cannot automatically tell. We only know that we should choose $\boldsymbol{v}^a$ so that $\boldsymbol{v}^a$ is zero as frequently as possible. By doing so, we eliminate convection essentially by definition, and we are left with a sub-stantially easier problem.

To see which reference velocity is easiest to use, we consider the diffusion apparatus shown in Fig. 3.1-2. This apparatus consists of two bulbs, each of which contains a gas or liquid solution of different composition. The two bulbs are connected by a long, thin capillary containing a stopcock. At time zero, the stopcock is opened; after an experi-mentally desired time, the stopcock is closed. The solutions in the two bulbs are then analyzed, and the concentrations are used to calculate the diffusion coefficient. The equations used in these calculations are identical with those used for the diaphragm cell.

Here, we examine this apparatus to elucidate the interaction of diffusion and convec-tion, not to measure the diffusion coefficient. The examination is easiest for the special cases of gases and liquids. For gases, we imagine that one bulb is filled with nitrogen and the other with hydrogen. During the experiment, the number of moles in the left bulb always equals the number of moles in the identical right bulb because isothermal and isobaric ideal gases have a constant number of moles per volume. The volume of the left bulb equals the volume of the right bulb because the bulbs are rigid. Thus the average velocity of the moles $\boldsymbol{v}^*$ and the average velocity of the volume $\boldsymbol{v}^0$ are both zero.

In contrast, the average velocity of the mass $\boldsymbol{v}$ in this system is not zero. To see why this is so, imagine balancing the apparatus on a knife edge. This edge will initially be located left of center, as in Fig. 3.1-2(b), because the nitrogen on the left is heavier than the hydrogen on the right. As the experiment proceeds, the knife edge must be shifted toward the center because the densities in the two bulbs will become more nearly equal.

Thus, in gases, the molar and volume average velocities are zero but the mass average velocity is not. Therefore, the molar and volume average velocities allow a simpler de-scription in gases than the mass average velocity.

We now turn to the special case of liquids, shown in Fig. 3.1-2(c). The volume of the solution is very nearly constant during diffusion, so that the volume average velocity is very nearly zero. This approximation holds whenever there is no significant volume change after mixing. In my experience, this is true except for some alcohol–water sys-tems, and even in those systems it is not a bad approximation.

[Figure]

Fig. 3.1-2. An example of reference velocities. Descriptions of diffusion imply reference to a velocity relative to the system's mass or volume. Whereas the mass usually has a nonzero velocity, the volume often shows no velocity. Hence diffusion is best referred to the volume's average velocity.

The other two velocities are more difficult to estimate. To estimate these velocities for one case, imagine allowing 50-weight percent glycerol to diffuse into water. The volume changes less than 0.1 percent during this mixing, so that the volume average velocity is very nearly zero. The glycerol solution has a density of about 1.1 $g/cm^3$, as compared with water at 1 $g/cm^3$, so that the mass density changes about 10 percent. In contrast, the glycerol solution has a molar density of about 33 mol/l, as compared with water at 55 mol/l; so the molar concentration changes about fifty percent. Thus the mass average velocity will be nearer to zero than the molar average velocity.

Thus in this set of experiments, the molar and volume average velocities are zero for ideal gases and the volume and mass average velocities are close to zero for liquids. The mass average velocity is often inappropriate for gases, and the molar average velocity is rarely used for liquids. The volume average velocity is appropriate most frequently, and so it will be emphasized in this book.

**3.2 Different Forms of the Diffusion Equation**

The five most common forms of diffusion equations are given in Table 3.2-1. Each of these forms uses a different way to separate diffusion and convection. Of course,

---

## Author Response (AR1)

**HESS manuscript:** https://doi.org/10.5194/egusphere-2024-2023

**Title:** Technical note: An assessment of the relative contribution of the Soret effect to open water evaporation

**Authors:** Roderick ML & Shakespeare CJ

**Combined response to the reviews following editorial decision of "minor revision". RC1 is at the end because it is the longest response.**

**General Overview of Revisions**

**Here we first summarise the main changes.**

**[Line numbers are those in the new manuscript labelled as V4]**

(a) **In response to RC5 we now refer to the Griffani et al 2024 paper in the introduction. On reflection of this comment, we have also added an additional empirical analysis for the thermal diffusion factor and this is included in new Appendix A and we have added relevant text to the manuscript (lines 144-154).**

(b) **In response to RC1 we have changed the basic equation of the analysis (see new Eq. 9, line 211) to include the requested advective flux and the total evaporation is now split into three components. We have included the mathematical derivation in a new Appendix B. This change required additional paragraphs throughout to consider the new flux within the body of the manuscript. The inclusion of this new flux did not materially change the conclusion of the manuscript.**

(c) **Because of (b) we decided to change our original Figure 2. The top panels from the old Figure 2 have been put into a new Appendix C. The bottom right panel of the old Figure 2 is now included in a new Figure 3. The new Figure 2 includes a plot of the three fluxes and the three relative fluxes are now shown as histograms.**

(d) **In response to RC3 we have conducted a more extensive search and identified the 2015 Persian Gulf heatwave as an example that has been included in new Appendix E and discussed in the main text in several places.**

**We believe these changes have improved the manuscript and we have noted this in the acknowledgements.**

**The detailed response to each reviewer comment is shown below.**

**Response to RC2 (Anon. Reviewer 2, 6 Sep 2024)**

*Review comments in italics.*

**Author Response in bold.**

*Roderick + Shakespeare*

1. *This is an unusual and nicely scholarly piece of work which should be published, more or less as is.*

**We thank the reviewer for the comments.**

2. *It is interesting to discover that the Soret effect has received so little experimental investigation. In that regard, I note that philosopher Nancy Cartwright in her book How the Laws of Physics Lie uses the Soret effect as an established example of a coupled-flux process in discussing causal inference. Perhaps less established than she thought.*

**As our references show, the Soret effect has been extensively investigated by physicists originally interested in developing/testing/refining the kinetic theory of gases primarily in the first half of the twentieth century. What the reviewer is referring to is that the Soret effect did not make it's way to other scientific disciplines (e.g. hydrology, atmospheric science, etc.).**

3. *I have really only two comments. The first is to wonder if the authors might spend a little more time in discussing the boundary layer structure in evaporation: how does the temperature vary across it? ... Can we assume that there is local kinetic (thermal) equilibrium within the boundary layer? What are reasonable boundary layer thicknesses and temperature gradients? There is brief mention only in lines 170-173.*

**As described in the manuscript, and implied by the equation, we have assumed the vapour is saturated at the liquid surface and is uniform above the boundary layer. Between the surface and top of the boundary layer we assume a linear profile as described in detail in the cited reference (Lim et al, 2012). This was beyond the scope of the article but we reproduce Fig. 1 from Lim et al (2012) below (see Fig. R1 below) to show that the details are available for an interested reader. A similar profile is used to model the temperature. Boundary layer thicknesses are also reported in Lim et al., and depend on the wind speed.**

**We have now added new text after Eqn 5 (lines 163-165) to explain this in more detail than we had originally provided.**

[Figure]

**Fig. 1.** A schematic diagram of the "threshold model" adopted here for the variation in vapour pressure ($e$) with height ($z$) above the evaporating surface. After Leighly (1937) and Machin (1964, 1970).

**Fig. R1 Reproduced from Lim et al (2012)**

4. *The second is to ask what is the connection between the the framework/ analysis set out in this paper and the description of thermal diffusion in porous media (water, liquid and vapour) originally set out by Philip and de Vries (1957) and later papers (perhaps Luikov too around the same time). Have I missed something here or should these analyses all be consistent?*

**Yes, they should all be consistent.**

**The Philip and de Vries (1957) work was dealing with a much more complex situation with solids as well as water in liquid and gas phases. If you look at their formulation, their original equation (reproduced here) for the mass flux of water vapor through the gas phase was,**

$$q_{\mathrm{vap}} = -D_{\mathrm{atm}}\nu\alpha a \nabla\rho \qquad (1)$$

**Their eqn 1 is actually based on Fick's Law (using vapor density $\nabla\rho$ as the driving force). In their Eqn 9 they used a classical 'Darcy' formulation for bulk flow of liquid.**

**No doubt one could reformulate the Philip and de Vries (1957) result in different ways but that is well beyond the scope of this paper. We note that the magnitude of thermodiffusion in soil (in either vapor or liquid phases) would be small as we have found here and could be ignored as Philip and de Vries have implicitly done.**

**Response to RC3 (Anon. Reviewer 3, 18 Sep 2024)**

*Review comments in italics.*

**Author Response in bold.**

1. *This manuscript gives a detailed discussion on the relative contributions of Fickian's diffusion and Soret effect on open water evaporation, and justifies the popular practice of estimating open water evaporation through the water vapor concentration gradients. The authors prove that the Soret effect is two orders of magnitude smaller than that by concentration-dependent diffusion. I consider such kind of work is just quite rare because of limited laboratory experiments and it is valuable for us to understand the processes behind it.*

**We thank the reviewer for the comments.**

2. *In figure1, the thermal diffusion factor is 0.05 for N2-N2O and N2-CO2, but 0.33 for H2-CO2. It seems that even though the thermal diffusion factor of 0.33 is used for Soret effect estimation, its contribution is still much smaller compared to the concentration gradients.*

**We assume the reviewer means that even if we used the (incorrect) larger value of 0.33 (instead of 0.05) the conclusions would not be materially altered. If so, we agree.**

3. *In abstract, the authors mentioned that "under typical environmental conditions it is at least two orders of magnitude smaller than classical concentration-dependent mass ('Fickian') diffusion.". From Figure1 and Figure 2, we could find that the Soret effect can be neglected in open water estimation. I 'm just wandering under what conditions in open water evaporation estimation (fresh water and saline water) the Soret effect can not be ignored? If no, please add some examples in the discussion of the manuscript.*

**We have revised the text by including a new example of an extreme case (2015 Persian Gulf Heatwave, see new Appendix D) and also substantially extended the text about this point (lines 237-265) and added a new paragraph in the Discussion (lines 308-327).**

4. *Further, the figure quality is not good, please make figures with good quality.*

**The manuscript was prepared using 'cut and paste' to insert the figures from a PDF into a Word document and we may have inadvertently used a low resolution during that process. However, for the final printed version the figures will be high quality PDFs. We have increased the font size on the axis labels.**

5.  *Generally, I consider the manuscript is a good materials that can help us to understand clear the water-atmosphere interaction processes.*

**We thank the reviewer for the comments.**

**Response to RC4 (Anon. Reviewer 4, 23 Sep 2024)**

*Review comments in italics.*

**Author Response in bold.**

1.  *This study quantifies the magnitude of the Soret effect on open water evaporation and demonstrate that it is typically two orders of magnitude smaller than the mass diffusion component (Fickian diffusion). This finding justifies the common practice of ignoring the Soret effect when describing evaporation in hydrological sciences.*

    *I believe this is an important study that should be accepted after minor corrections and clarifications. The manuscript is well-written and exhibits excellent readability. However, it may benefit from clarifications regarding the following points.*

**We thank the reviewer for the thoughtful comments.**

*Comments:*

2.  *C1: Multiple times throughout the manuscript, the sentences give an impression that evaporation is entirely a Fick's diffusion process (Line 26,212). However, estimation of evaporation also requires an explicit consideration of an energy term. Over open-water surfaces the gradient in the water-vapor is further strongly controlled by changes in temperature and incoming energy as reflected in the classical equilibrium energy partitioning approach (Slatyer and McIlroy, 1961).*

**The energy balance approach (e.g. Slatyer & McIllroy 1961) is an alternative to the mass transfer approach (e.g. Fick's law) and this has been widely used in hydrology and agriculture (also see Yang & Roderick 2019, QJRMS, 145, 1118-1129). However, it has been held for more than 200 years (since Dalton's 1802 paper) that evaporation can be specified solely using a mass transfer approach.**

**Energetic constraints are needed in addition to (but do not replace) the mass transfer formulation for evaporation if one seeks to model the evolution of the system over time (e.g., in a land surface, atmosphere or ocean model). Here, however, we have direct experimental control of all key variables (air temperature/humidity and windspeed) and measure evaporation and surface temperature directly which avoids the need for such modelling.**

*Review comments in italics.*

**Author Response in bold.**

1.  *This short technical note deals with quantifying the relative contribution of the Soret effect (i.e. thermodiffusion) compared to regular "Fickian" diffusion in the evaporation of open water bodies. The motivation behind the paper is to justify the standard practice of neglecting the Sorret effect for evaporation. For this purpose, it is based on the gas kinetic theory of Chapman and his colleagues (completed with experimental determination of the Sorret effect in binary mixtures) and on a recent dataset of evaporation rates under controlled conditions.*

    *I think the paper is of general interest for people working of water vapor transport (even beyond the sole problem of open water evaporation) and is well suited for HESS. There is however one main point of concern (General Comment 1) that I would like the authors to clarify.*

**We thank the reviewer for the comments.**

2.  *General Comments 1 – It is stated at the very start of the paper that the evaporation rate of an open water body is controlled by the ability of water vapor to diffuse in the air. From what I understand this is clearly the case when the air above the water is still (in this case, diffusion in the limiting factor, effectively setting its rate for the evaporation). However, as soon as wind and turbulence is involved, I have issues understanding to what extent the evaporation rate remains controlled by the molecular diffusion in the air. I'm no boundary-layer scientist, but from what I understand, the diffusion flux of Eq. 4 applies within the micro layer (following the wording of Roland B. Stull's "An introduction to Boundary Layer Meteorology"), i.e. the zone just above the surface where molecular diffusion dominates. Thus, if it is to be translated into Eq. 5 I would say that the gradients of concentration and temperature are to be taken across this micro-layer. And I'm not sure that the concentration and temperature at the top of this micro-layer can be taken as xa and Ta (as they are influenced by the surface). Otherwise, if xa and Ta are taken to define the gradients, I think the diffusivity should rather be some "effective" diffusivity (including turbulent effects) and thus does not match the molecular diffusivity anymore. And in this case, it is not clear to me that one can upscale the Sorret effect to the whole boundary layer in a similar fashion (i.e. that there is an effective Sorret flux, including turbulence, that has the same form and the same thermal diffusivity ratio as in the purely molecular case). In other words, I think the problem boils down to the difficulty of reconstructing the surface concentration and temperature gradients based on the "air" values, which I assume can be significantly different from what happen in the micro-layer. I would thus like the authors to clarify this point. Especially, references to pre-existing literature treating this problem and relating molecular diffusion in the micro-layer to the air temperature/concentration would be appreciated.*

**We thank the reviewer for this comment. Please see the response to RC3, comment 3.**

3. *2 – From what I understand the motivation behind this close look at the Sorret effect stems from the recent study of Griffani et al. (2024), that states that thermo-diffusion can be an effective mechanism of water vapor transport and should not always be neglected. However, this motivation only appears in the discussion. I think it could be quite beneficial to include this in the introduction, as it relates to the state-of-the-art on the subject.*

**We accept the point and we have now added a reference to Griffani et al (2024) as well as the necessary text in the introduction as requested.**

*Specific and technical comments*

4. *Abstract – I would systematically say "evaporation from open water" rather than simply "evaporation".*

**Done.**

5. *L15 – As mentioned in the General Comment 1, references to pre-existing literature would be beneficial here.*

**We have added a reference to the classical Monteith and Unsworth 2008 textbook at the end of the first sentence as requested.**

6. *L21 – The mention of the Duffour effect is a bit off to me, especially as it is no longer mentioned in the text. It could potentially be discussed a bit more in the Discussion and Conclusions section, notably mentioning that the Onsager reciprocal relations allows one to estimate the Duffour effect from Sorret.*

**We agree that the Dufour effect is left hanging on it's own. We just wanted to make the point that the Onsager-based coupling leads to other effects. We have rewritten the text (in paragraph 1) to highlight the symmetrical nature of the Onsager-based flux-coupling to give the necessary context that we believe was missing from the original manuscript.**

7. *L76 and L84 - I would say "limiting conditions" rather than "boundary conditions".*

**Done.**

8. *L82 – If I'm not wrong the quadratic form was proposed in the work of Chapman (and certainly others). You could refer to them to justify this specific functional form*

**We have cited the first standard text on the topic (Grew and Ibbs 1952).**

9. *Eq 5 - It might be just me, but I'm not fond of mixing alphabetical and digits in Equations, as it obscure physical variables from actual math constant (and I find it harder to read and interpret). I would keep alpha_T rather than 0.05.*

**We accept the point that this is a matter of "taste". We also prefer using symbols but in this case we have supplied the actual numeric value that is assumed unchanging.**

*10. L136 – It relates to General comment 1. Could you elaborate on the physical significance of Delta z? Is it the thickness of the micro-layer (where diffusion dominates)?*

**Yes, it is the thickness you mentioned. We have modified the text after Eqn 5 to explain this more fully (also see RC2, comment 3: RC5, comment 2).**

*11. Eqs 6a and 6b – Same as Eq. 5. I would go straight to the point and say that Eq 5 using some standards values yields a 99.6%/0.4% partition for the Fickian and Sorret fluxes.*

**Again a matter of taste. We think that having the numbers written out makes it very clear that the Soret effect will be small regardless of the boundary layer model chosen.**

*12. L149 – For me, "vanishingly small" implies that the Sorret contribution strictly goes to zero when xs equals xa. However, I do not think it's the case as small Sorret contribution remains non zero (if xs is different from 0 or 1, and Ts different from Ta).*

**We have modified the text accordingly.**

*13. L155 – If it's the air temperature, please use Ta rather than T.*

**Done.**

*14. Figure 2 – I find it hard to determine the relative contribution of the Sorret flux in panel c near the origin of the graph. Perhaps add a second y-scale the relative contribution of the Sorret flux as a scatter of the total evaporation flux.*

**We agree that the Soret flux is hard to precisely determine near the origin but the main point here is not the exact numerical value, but rather that it is very small.**

*15. L194 – I think the wording could be improved. From what I understand, the issue is that Griffani et al.'s work is based on Landau's derivation which assumes that (ii) the water vapor molecule are much lighter than the dry air and (ii) the collisions are elastic; both assumption not applying to the actual mixture of water and dry air. The current wording rather suggests that the issues are (i) Griffani et al. is based on Landau (M=1 and elastic collisions) and (ii) the collisions are assumed elastic; in which the same issue actual appears twice.*

**We have completely rewritten this paragraph (lines 295-306) and we believe the new wording satisfies this suggestion.**

*16. L209 – Taking alpha_t as 0.05 rather than 0.5 effectively reduces the Sorret flux by an order of magnitude, and the 30% contribution of Sorret in Griffani et al. now becomes a 3% contribution, which is deemed negligible. That being said, I would still leave the door open that in almost all natural conditions, the Sorret effect appears negligible, but that pathological cases (i.e. very high temperature gradients with little concentration gradients) can still exist (and it links with the paragraph below).*

**Reviewer 3 raised a similar sentiment (see point 3 in the response to Reviewer 3).**

**We have now modified the text (lines 237-265) to include a more extensive analysis, and we have added an example (Persian Gulf Heatwave, Appendix D). This point is further pursued in the discussion with a new paragraph (lines 308-327) describing this important point.**

**Response to RC1 (Dr Andrew Kowalski, 3 Sep 2024)**

*Review comments in italics.*

**Author Response in bold.**

**Given the nature of the review we have chosen to reverse the order of the questions to be addressed.**

*2.      Independent of this, I point out that the authors' Eq. (1) is dimensionally inhomogeneous unless the diffusive flux density (J) is specified in molar terms, with units as in Table 1 rather than the mass-based units that they indicate at line 62. Also, the axis labels should be larger in order to be legible, particularly for Figure 2.*

**Yes, there was a 'typo' on line 62 where the units should have been molar.**

**We have made the text larger on the axes of both Figures 1 and 2 as suggested.**

**Thank you.**

*1. The manuscript by Roderick and Shakespeare purports to characterise the influence of the Soret effect, whereby temperature gradients influence mass diffusion, versus the classical concentration-dependent mass ('Fickian') diffusion. But in order to do this requires first correctly characterising Fickian diffusion, and this I believe the authors have not yet done. In brief, the authors have specified Fick's law based on gradients in the molar fraction, whereas Newtonian analyses demonstrate that it must be specified in terms of the mass fraction, and the difference between the two is hardly trivial for fluids of varying molecular mass. Respectfully, I therefore believe that the manuscript should be*

*rejected. My arguments for why their specification of Fick's 1ˢᵗ Law is incorrect are laid out in an open-access paper (Kowalski et al., 2021) that can be accessed here (https://link.springer.com/article/10.1007/s10546-021-00605-5; see sections 3.2 and 4 in particular), but are reinforced in the attached PDF file.*

**We thank the reviewer for their considered response and for the additional document (i.e., the uploaded PDF) based on "Isaac" which we enjoyed reading. We have also carefully read the cited 2021 Boundary Layer Meteorology article as well as the earlier 2017 article in AtmChemPhys.**

**We were somewhat surprised by the ferocity (i.e., recommend complete rejection) given the highly favourable comments by the other six reviewers of the manuscript. With that background we have taken the comment seriously. In fact this was the first comment posted on our manuscript (on 3 Sep 2024) and one of us (MLR) has spent nearly all of the available time since then undertaking additional reading/research to seriously address the comment. Our combined knowledge of "diffusion" has increased substantially and for that we thank the reviewer. The extra work was worth it in this case.**

**The underlying basis of the assertion that we MUST use a mass-based framework has two separate points. The first is simply that one has to use a mass-based framework and the second is that we have ignored a small advective flux (i.e., the so-called Stefan flow). For the latter we accept the criticism and we have now included the advective flux (see new Eq. 9, line 211) and a justification for this new flux (lines 194-211) has now been added. In addition we have included the mathematical derivation of the new Eq. 9 in Appendix B and modified the discussion (lines 329-333).**

**In implementing the new flux we have followed the standard engineering literature and cited two main references who show that one can use a molar or mass-based analysis (Kreith, Cussler).**

**We note that there is a vast scientific literature (e.g. Cullinan 1965; Brady 1975; Miller 1986) that establishes the complete equivalence of mass- and molar-based frameworks for describing diffusion. We also identified a very succinct and elegant tabular summary of this very point in the famous Cussler textbook on diffusion (Cussler 2009: Table 3.2-1) that is reproduced below in Fig. R2. Interestingly, in Hydrology (and other climate related fields) we usually specify the diffusion coefficient of water in air based on laboratory experiments. It turns out that those experimental results are actually based on a volume-based reference velocity as is most gas-based analysis and that most closely maps to a molar average velocity.**

Table 3.2-1 *Different forms of the diffusion equation*

| Choice | Total flux (diffusion + convection) | Diffusion equation | Reference velocity | Where best used |
|---|---|---|---|---|
| Mass | $n_1 = j_1^m + \rho_1 v$ | $j_1^m = \rho_1(v_1 - v)$
 $= -D\rho \nabla \omega_1$ | $v = \omega_1 v_1 + \omega_2 v_2$
 $\rho v = n_1 + n_2$ | Constant-density liquids; coupled mass and momentum transport |
| Molar | $n_1 = j_1^* + c_1 v^*$ | $j_1^* = c_1(v_1 - v^*)$
 $= -Dc\nabla y_1$ | $v^* = y_1 v_1 + y_2 v_2$
 $cv^* = n_1 + n_2$ | Ideal gases where the total molar concentration $c$ is constant |
| Volume | $n_1 = j_1 + c_1 v^0$ | $j_1 = c_1(v_1 - v^0)$
 $= -D\nabla c_1$ | $v^0 = c_1 \bar{V}_1 v_1 + c_2 \bar{V}_2 v_2$
 $= \bar{V}_1 n_1 + \bar{V}_2 n_2$ | Best overall; good for constant-density liquids and for ideal gases; may use either mass or mole concentration |
| Solvent | $n_1 = j_1^{(2)} + c_1 v_2$ | $j_1^{(2)} = c_1(v_1 - v_2)$
 $= -D_1 \nabla c_1$ | $v_2$ | Rare except for membranes; note that $D_1 \neq D_2 \neq D$ |
| Maxwell–Stefan | | $\nabla y_1 = \dfrac{y_1 y_2}{D'}(v_2 - v_1)$ | None | Written for ideal gases; difficult to use in practice |

**Figure R2**      **Reproduction of Table 3.2-1 from Cussler (2009: p. 60) showing the equivalence of different forms of the diffusion equation.**

**Finally, we refer back to the 'Isaac' examples provided by the reviewer. These involve two different gases (of different molecular mass) and what happens when they mix. This problem has been explicitly dealt with by Cussler (2009: p. 58-59) and Cussler agrees with the reviewer that that centre of mass will change. But Cussler also points out that the volume (and mole) average velocity is zero and so this is the easiest diffusion equation to use and is recommended for that reason which refutes the assertion by the reviewer. We have included "snapshots" from p. 58-59 of Cussler in an appendix to this response.**

***References***

**Brady, J. B.: Reference frames and diffusion coefficients, American Journal of Science, 275, 954-983, doi.org/10.2475/ajs.275.8.954, 1975.**

**Cullinan, H. T., Jr.: Analysis of Flux Equations of Multicomponent Diffusion, Industrial & Engineering Chemistry Fundamentals, 4, 133-139, 10.1021/i160014a005, 1965.**

Cussler, E. L.: Diffusion: Mass Transfer in Fluid Systems, 3$^{rd}$ ed., Cambridge University Press, Cambridge, UK, 2009.

Kowalski, A. S., Serrano-Ortiz, P., Miranda-García, G., and Fratini, G., 2021. "Disentangling turbulent gas diffusion from non-diffusive transport in the boundary layer." Boundary-Layer Meteorology, **179** (3), 347-367.

Kowalski, A. S.: The boundary condition for vertical velocity and its interdependence with surface gas exchange, Atmos. Chem. Phys., 17, 8177-8187, 10.5194/acp-17-8177-2017, 2017.

Kreith, F., Boehm, R., Raithby, G., Hollands, K., Suryanarayana, N., Modest, M., VP, V., Chen, J., Lior, N., Shah, R., Bell, K., Moffat, R., Mills, A., Bergles, A., Swanson, L., Antonetti, V., Irvine Jr, T., and Capobianchi, M.: Heat and Mass Transfer, in: Mechanical Engineering Handbook, edited by: Kreith, F., CRC Press LLC, Boca Raton, 1999.

Miller, D. G.: Some comments on multicomponent diffusion: negative main term diffusion coefficients, second law constraints, solvent choices, and reference frame transformations, The Journal of Physical Chemistry, 90, 1509-1519, 10.1021/j100399a010, 1986.

**Appendix – Snapshots of pages 58 and 59 from Cussler (2009)**

In more exact terms, we define the total mass flux $n_1$ as the mass transported per area per time relative to fixed coordinates. This flux, in turn, is used to define an average solute velocity $v_1$:

$$n_1 = c_1 v_1 \tag{3.1-2}$$

where $c_1$ is the local concentration. We then divide $v_1$ into two parts:

$$n_1 = c_1(v_1 - v^a) + c_1 v^a = j_1^a + c_1 v^a \tag{3.1-3}$$

where $v^a$ is some convective "reference" velocity. The first term $j_1^a$ on the right-hand side of this equation represents the diffusion flux, and the second term $c_1 v^a$ describes the convection.

Interestingly, there is no clear choice for what this convective reference velocity should be. It might be the mass average velocity that is basic to the equations of motion, which in turn are a generalization of Newton's second law. It might be the velocity of the solvent, because that species is usually present in excess. We cannot automatically tell. We only know that we should choose $v^a$ so that $v^a$ is zero as frequently as possible. By doing so, we eliminate convection essentially by definition, and we are left with a substantially easier problem.

To see which reference velocity is easiest to use, we consider the diffusion apparatus shown in Fig. 3.1-2. This apparatus consists of two bulbs, each of which contains a gas or liquid solution of different composition. The two bulbs are connected by a long, thin capillary containing a stopcock. At time zero, the stopcock is opened; after an experimentally desired time, the stopcock is closed. The solutions in the two bulbs are then analyzed, and the concentrations are used to calculate the diffusion coefficient. The equations used in these calculations are identical with those used for the diaphragm cell.

Here, we examine this apparatus to elucidate the interaction of diffusion and convection, not to measure the diffusion coefficient. The examination is easiest for the special cases of gases and liquids. For gases, we imagine that one bulb is filled with nitrogen and the other with hydrogen. During the experiment, the number of moles in the left bulb always equals the number of moles in the identical right bulb because isothermal and isobaric ideal gases have a constant number of moles per volume. The volume of the left bulb equals the volume of the right bulb because the bulbs are rigid. Thus the average velocity of the moles $v^*$ and the average velocity of the volume $v^0$ are both zero.

In contrast, the average velocity of the mass $v$ in this system is not zero. To see why this is so, imagine balancing the apparatus on a knife edge. This edge will initially be located left of center, as in Fig. 3.1-2(b), because the nitrogen on the left is heavier than the hydrogen on the right. As the experiment proceeds, the knife edge must be shifted toward the center because the densities in the two bulbs will become more nearly equal.

Thus, in gases, the molar and volume average velocities are zero but the mass average velocity is not. Therefore, the molar and volume average velocities allow a simpler description in gases than the mass average velocity.

We now turn to the special case of liquids, shown in Fig. 3.1-2(c). The volume of the solution is very nearly constant during diffusion, so that the volume average velocity is very nearly zero. This approximation holds whenever there is no significant volume change after mixing. In my experience, this is true except for some alcohol–water systems, and even in those systems it is not a bad approximation.

[Figure]

(a) Basic apparatus

Long, thin capillary gives resistance to diffusion

Stopcock for starting and stopping experiment

Two identical bulbs stirred by free convection

(b) Gas example

N₂   H₂

Initial center of mass   Final center of mass

(c) Liquid example

Glycerol   Water

Initial center of mass   Final center of mass

Fig. 3.1-2. An example of reference velocities. Descriptions of diffusion imply reference to a velocity relative to the system's mass or volume. Whereas the mass usually has a nonzero velocity, the volume often shows no velocity. Hence diffusion is best referred to the volume's average velocity.

The other two velocities are more difficult to estimate. To estimate these velocities for one case, imagine allowing 50-weight percent glycerol to diffuse into water. The volume changes less than 0.1 percent during this mixing, so that the volume average velocity is very nearly zero. The glycerol solution has a density of about 1.1 g/cm$^3$, as compared with water at 1 g/cm$^3$, so that the mass density changes about 10 percent. In contrast, the glycerol solution has a molar density of about 33 mol/l, as compared with water at 55 mol/l; so the molar concentration changes about fifty percent. Thus the mass average velocity will be nearer to zero than the molar average velocity.

Thus in this set of experiments, the molar and volume average velocities are zero for ideal gases and the volume and mass average velocities are close to zero for liquids. The mass average velocity is often inappropriate for gases, and the molar average velocity is rarely used for liquids. The volume average velocity is appropriate most frequently, and so it will be emphasized in this book.

**3.2 Different Forms of the Diffusion Equation**

The five most common forms of diffusion equations are given in Table 3.2-1. Each of these forms uses a different way to separate diffusion and convection. Of course,